# Learning Optimal Multimodal Information Bottleneck Representations

Qilong Wu [* 1]   Yiyang Shao [2]   Jun Wang [* 3]   Xiaobo Sun [* 4]

## Abstract

Leveraging high-quality joint representations from multimodal data can greatly enhance model performance in various machine-learning based applications. Recent multimodal learning methods, based on the multimodal information bottleneck (MIB) principle, aim to generate optimal MIB with maximal task-relevant information and minimal superfluous information via regularization. However, these methods often set *ad hoc* regularization weights and overlook imbalanced task-relevant information across modalities, limiting their ability to achieve optimal MIB. To address this gap, we propose a novel multimodal learning framework, Optimal Multimodal Information Bottleneck (OMIB), whose optimization objective guarantees the achievability of optimal MIB by setting the regularization weight within a theoretically derived bound. OMIB further addresses imbalanced task-relevant information by dynamically adjusting regularization weights per modality, promoting the inclusion of all task-relevant information. Moreover, we establish a solid information-theoretical foundation for OMIB's optimization and implement it under the variational approximation framework for computational efficiency. Finally, we empirically validate the OMIB's theoretical properties on synthetic data and demonstrate its superiority over the state-of-the-art benchmark methods in various downstream tasks.

## 1. Introduction

In the parable "*Blind men and an elephant*", a group of blind men attempts to perceive the elephant's shape through touch,

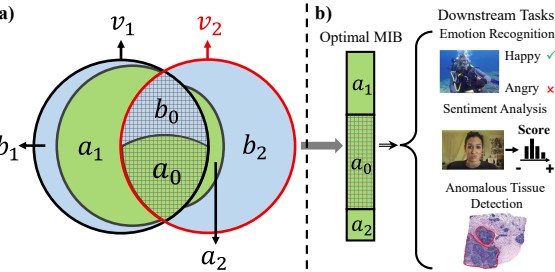

Figure 1: **a)** Venn diagrams for two data modalities ($v_1$ and $v_2$). The gridded area represents consistent information, while the non-gridded area denotes modality-specific information. Task-relevant information is highlighted in green, whereas superfluous information is shown in blue. **b)** An optimal MIB should exclusively contain task-relevant, non-superfluous information (i.e., $a_0, a_1$ and $a_2$) to be utilized in downstream tasks for enhanced performance.

but each inspects only a single, distinct part (e.g., tusk, leg). Consequently, they deliver conflicting descriptions, as their judgments are based solely on the part they touch.

In the context of machine learning, this parable underscores the significance of multimodal learning, which integrates and leverages multimodal data (akin to the elephant's body parts) to grasp a holistic understanding, thereby enhancing inference and prediction accuracy. In multimodal learning, unimodal features are extracted from each modalities and fused with various fusion strategies, such as tensor-based (Zadeh et al., 2017; Liu et al., 2018), attention-based (Guo et al., 2020; Xiao et al., 2020; Zhang et al., 2023), and graph-based (Arun et al., 2022; Huang et al., 2021), to generate multimodal embeddings. However, a major limitation of these methods is their potential to include superfluous and redundant information from each modality, increasing embedding complexity and the risk of overfitting (Mai et al., 2023; Wan et al., 2021).

From an information theory perspective, a comprehensive multimodal learning method should account for five factors: **consistency** (Tian et al., 2021), **specificity** (Liu et al., 2024), **complementarity** (Wan et al., 2021), **sufficiency** (Federici et al., 2020), and **conciseness** (a.k.a. nonredundancy) (Wang et al., 2019). As illustrated in Figure 1a, on the input side, *consistency* describes information shared

---

*Equal contribution  [1]School of Statistics and Mathematics, Zhongnan University of Economics and Law  [2]School of Finance, Zhongnan University of Economics and Law  [3]iWudao  [4]School of Medicine, Department of Human Genetics, Emory University. Correspondence to: Xiaobo Sun <xsun28@emory.edu>.

*Proceedings of the $42^{nd}$ International Conference on Machine Learning*, Vancouver, Canada. PMLR 267, 2025. Copyright 2025 by the author(s).

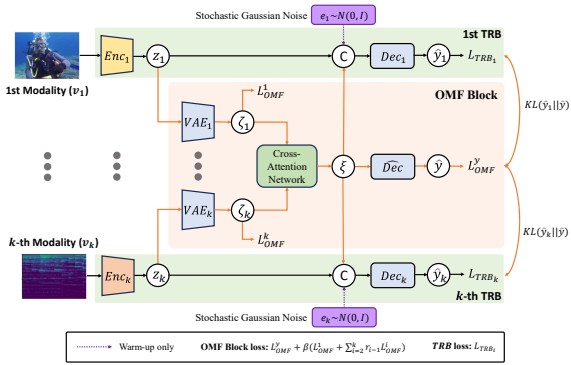

Figure 2: **OMIB Framework.** Here, 'C' represents the concatenation operation. For the definitions of other notations, refer to the Section 4 and Table 1.

across input modalities (gridded area), while *specificity* refers to the information unique to individual modalities (non-gridded area). Both consistent and modality-specific information may contain task-relevant (green area) or superfluous (gray area) components. *Complementarity* pertains to modality-specific, task-relevant information ($a_1$ and $a_2$), enabling multimodal embeddings to surpass unimodal ones in downstream tasks. On the output side, an optimal multimodal embedding (as shown in Figure 1b and Definition 5.4 below) must be *sufficient*, capturing maximal task-relevant information—both consistent ($a_0$) and complementary ($a_1, a_2$)—across modalities. Meanwhile, it should be *concise*, minimizing both cross-modality ($b_0$) and modality-specific superfluous information ($b_1, b_2$) to reduce complexity. This optimal multimodal embedding can then be applied to various downstream tasks, such as multimodal sentiment analysis (Mai et al., 2023) and pathological tissue detection using histology and gene expression data (Xu et al., 2024b)), for enhanced performance.

To this end, multimodal learning methods based on the Multimodal Information Bottleneck (MIB) principle have been developed, which generally follow a common paradigm: modality-specific representations are extracted and fused into MIB via deep networks. The MIB is then optimized to balance two objectives: maximizing mutual information between the embeddings and task-relevant labels for sufficiency; and minimizing mutual information between the embeddings and the raw input to purge superfluous information and ensure conciseness (Wang et al., 2019; Wan et al., 2021; Zhang et al., 2023; Fang et al., 2024). This process is formalized as:

$$
\begin{aligned}
x_i &= E_i(v_i), \\
z &= F(x_1, x_2, ...), \\
\mathcal{O}(v_i, z, y) &:= \max_z I(y; z) - \sum_i \beta_i I(v_i; z),
\end{aligned}
\tag{1}
$$

where $E_i, F, I$, and $\mathcal{O}$ represent the modality-specific encoder, multimodal fusion function, mutual information function, and optimization objective, respectively. $v_i, x_i, z$, and $y$ denote the raw data of the $i$-th modality, its extracted representation, the MIB, and task labels. Particularly, $\beta_i$ serves as the regularization parameter for constraining superfluous information between the MIB and the $i$-th modality.

Despite their promising performance, these methods face three major limitations. First, the achievability of optimal MIB is not guaranteed. Since the regularization parameters (e.g., $\beta$s in Equation (1)) control the trade-off between sufficiency and conciseness, their values are critical for optimizing MIB (Tian et al., 2021). If the value is too small, superfluous information may be retained, leading to suboptimal MIB. Conversely, if too large, task-relevant information may be excluded due to an overemphasis on conciseness, compromising MIB's sufficiency. However, existing MIB methods determine these parameter values in an *ad hoc* manner, limiting their ability to achieve an optimal MIB. Second, an ideal MIB method should dynamically adjust regularization weights based on the remaining task-relevant information in each modality. Specifically, a modality should receive a lower regularization weight if a significant portion of its task-relevant information is left out of the MIB, and vice versa. However, existing MIB methods typically assign fixed, *ad hoc* regularization weights to each modality during training. When task-relevant information is imbalanced across modalities, some modalities may contain minor but crucial task-relevant information (e.g., $a_2$ in $v_2$ in Figure 1). If such a modality is assigned an excessively large regularization weight, its task-relevant information may be inadvertently excluded from the MIB (Fan et al., 2023). Finally, these methods lack theoretical comprehensiveness, as they either fail to incorporate all five aforementioned factors into the optimization objective or do not acknowledge their distinct roles in guiding optimization. For instance, the study in (Tian et al., 2021) overlooks complementarity, while CMIB-Nets (Wan et al., 2021) does not account for consistent, superfluous information. Additionally, in the theoretical analyses of methods such as (Fang et al., 2024; Wan et al., 2021), the two types of task relevant information—consistent (e.g., $a_0$ in Figure 1) and modality-specific (e.g., $a_1, a_2$)— are not distinguished, despite their differing impacts on the optimization objective.

To address these issues, we propose a novel MIB-based multimodal learning framework, termed **Optimal Multimodal Information Bottleneck (OMIB)**, to learn task-relevant optimal MIB representations from multi-modal data for enhanced downstream task performance. OMIB features theoretically grounded optimization objectives, explicitly linked to the dynamics of all five information-theoretical factors during optimization, ensuring a holistic and rigorous optimization framework. As illustrated in Figure 2,

OMIB comprises two components, including task relevance branches (TRBs) that extract sufficient representations from individual modalities, and an optimal multimodal fusion block (OMF), where modality-specific representations are fused by a cross-attention network (CAN) into MIB and optimized using a computationally efficient variational approximation (Alemi et al., 2017). Adhering to the MIB principle, the OMF block maximizes sufficiency while minimizing redundancy in the MIB. Particularly, by setting the redundancy regularization parameter in OMIB's objective function within a theoretically derived bound, OMIB guarantees the achievability of optimal MIB upon convergence of the OMF block training. Furthermore, our approach dynamically refines regularization weights per modality *as per* the distribution of their remaining task-relevant information. In summary, our contributions include:

- We propose OMIB, a novel framework for learning optimal MIB representations from multimodal data, with an explicit solution to address imbalanced task-relevant information across modalities.

- We provide a rigorous theoretical foundation that underpins OMIB's optimization procedure, establishing a clear connection between its objectives and the five information-theoretical factors: sufficiency, consistency, redundancy, complementarity, and specificity.

- We mathematically derive the conditions for achieving optimal MIB, marking, to our knowledge, the first endeavor in proving its achievability under the MIB principle.

- We validate OMIB's effectiveness on synthetic data and demonstrate its superiority over state-of-the-art MIB methods in downstream tasks such as sentiment analysis, emotion recognition, and anomalous tissue detection across diverse real-world datasets.

## 2. Related Work

### 2.1. Multimodal Fusion

Multimodal fusion methods can be categorized according to the fusion stage and techniques. Architecturally, fusion can happen at three stages: (1) Early fusion, which combines data at the feature level (Snoek et al., 2005), (2) Late fusion, integrating data at the decision level (Morvant et al., 2014), and (3) Middle fusion, where data is fused at intermediate layers to allow early layers to specialize in learning unimodal patterns (Nagrani et al., 2021). From the technique perspective, fusion approaches include: (1) Operation-based, combining features through arithmetic operations (El-Sappagh et al., 2020; Lu et al., 2021), (2) Attention-based, using cross-modal attention to learn interaction weights (Schulz et al., 2021; Cai et al., 2023), (3)

Tensor-based, modeling high-order interactions (Chen et al., 2020; Zadeh et al., 2017), (4) Subspace-based, projecting modalities into shared latent spaces (Yao et al., 2017; Zhou et al., 2021), and (5) Graph-based, representing modalities as graph nodes and edges (Parisot et al., 2018; Cao et al., 2021). In addition, recent studies also discuss the issue of modality imbalance, where strong modalities tend to dominate the learning process, while weak modalities are often suppressed (Peng et al., 2022; Zhang et al., 2024). Though effective, these methods typically fail to account for superfluous information and thus are prone to overfitting and sensitive to noisy modalities, limiting their practical robustness (Fang et al., 2024). MIB addresses these challenges by preserving task-relevant information while minimizing redundant content in the generated multimodal representations.

### 2.2. Multimodal Information Bottleneck

The Information Bottleneck (IB) framework (Tishby et al., 2000) provides a principled approach for learning compressed, task-relevant representations. It was first applied to deep learning by (Tishby & Zaslavsky, 2015) and later extended through the Variational Information Bottleneck (VIB) (Alemi et al., 2017), which employs stochastic variational inference for efficient approximations. Recently, IB has been adapted to more complex settings, such as multi-view (Wang et al., 2019; Federici et al., 2020) and multimodal learning (Tian et al., 2021). For example, L-MIB, E-MIB, and C-MIB (Mai et al., 2023) aim to learn effective multimodal representations by maximizing task-relevant mutual information, eliminating redundancy, and filtering noise, while exploring how MIB performs at different fusion stages. Secondly, MMIB-Zhang (Zhang et al., 2022) improves multimodal representation learning by imposing mutual information constraints between modality pairs, enhancing the model's ability to retain relevant information. Additionally, DISENTANGLEDSSL (Wang et al., 2024) relaxes the restrictions on achieving minimal sufficient information, thereby enabling the disentanglement of modality-shared and modality-specific information and enhancing interpretability. Lastly, DMIB (Fang et al., 2024) filters irrelevant information and noise, employing a sufficiency loss to preserve task-relevant data, ensuring robustness in noisy and redundant environments.

However, these methods often rely on *ad hoc* regularization weights and overlook the imbalance of task-relevant information across modalities, limiting their ability to fully optimize the MIB framework.

## 3. Notations

Here, we list the mathematical notations (Table 1) used in this study.

Table 1: Summary of notation.

| Notation | Description |
|---|---|
| $y$ | Task-relevant label. |
| $v_i$ | The $i$-th modality. |
| $z_i$ | The sufficient encoding of $v_i$ for $y$. |
| $\xi$ | MIB encoding. |
| $N$ | The total number of observations. |
| $H(*)$ | The entropy of variable $*$. |
| $F(*)$ | The information set inherent to variable $*$ (i.e., $F(x) = H(x)$). |
| $I$ | The mutual information function. |

## 4. Method

To clearly illustrate OMIB's framework, we start with the case of two data modalities (e.g., $v_1$ and $v_2$ in Figure 2), which can be readily extended to multiple modalities by adding additional modality branches (see Appendix D.1). We also provide a rigorous theoretical foundation for our methodology in Section 5.

**Warm-up training.** This phase consists of two task relevance branches (*TRB*) corresponding to $v_1$ and $v_2$, respectively. In the $i$-th TRB, $v_i$ is first encoded into a sufficient representation $z_i \in \mathbb{R}^d$ for task-relevant labels $y$:

$$z_i = Enc_i(v_i; \theta_{Enc_i}), s.t. I(z_i; y) = I(v_i; y), \quad (2)$$

where $Enc_i$ is an encoder, $\theta_{Enc_i}$ denotes its parameters. To ensure maximal sufficiency of $z_i$ for $y$, we concatenate it with a stochastic Gaussian noise, $e_i \in \mathbb{R}^k = N(0, I)$, before feeding it to a task-relevant prediction head $Dec_i$ (see Appendix H) to yield the predicted output $\hat{y}_i$:

$$\hat{y}_i = Dec_i([z_i, e_i]) \quad (3)$$

Via this step, $Enc_i$ is optimized to extract maximal task-relevant information from $v_i$, as it requires a higher signal-to-noise ratio in $z_i$ to yield accurate prediction from its corrupted version. The loss function for updating $Enc_i$ and $Dec_i$ is:

$$L_{TRB_i} = E_{v_i}[-\log p(\hat{y}_i | z_i, e_i)]$$
$$= -\frac{1}{N} \sum_{n=1}^{N} \log p(\hat{y}_i^n | z_i^n, e_i^n). \quad (4)$$

Note that the implementation of $\log p(\hat{y}_i | z_i, e_i)$ is task-specific. For classification tasks, it is implemented as $CE(\hat{y}_i || y)$, where $CE$ is the cross-entropy function; for SVDD-based anomaly detection (Ruff et al., 2018), it is $||\hat{y}_i - c||$, where $c$ is the unit sphere center of normal observations (see Appendix H); for regression tasks, it is $-||\hat{y}_i - y||$. The algorithmic workflow of the warm-up training is described in Appendix L.

**Main Training.** After the warm-up training, the model proceeds to main training, which includes an optimal multi-modal fusion (*OMF*) block in addition to the TRBs. In the OMF, $z_i, \forall i \in \{1, 2\}$, is used to generate the mean $\mu_i \in \mathbb{R}^k$ and variance $\Sigma_i \in \mathbb{R}^{k \times k}$ of a Gaussian distribution using a variational autoencoder ($VAE_i$):

$$\mu_i, \Sigma_i = VAE_i(z_i, \theta_{VAE_i}), \quad (5)$$

where $\theta_{VAE_i}$ represents the parameters of $VAE_i$. For efficient training and direct gradient backpropagation, the reparameterization trick (Kingma, 2013) is applied to generate $\zeta_i \in \mathbb{R}^k$:

$$\zeta_i = \mu_i + \Sigma_i \times \epsilon_i, \text{ where } \epsilon_i \sim N(0, I). \quad (6)$$

$\zeta_1$ and $\zeta_2$ are fused using a Cross-Attention Network (CAN) (Vaswani et al., 2017), whose architecture is detailed in Appendix H:

$$\xi = CAN(\zeta_1, \zeta_2, \theta_{CAN}), \quad (7)$$

where $\xi$ is the post-fusion embedding, which is then passed to a task-relevant prediction head $\widehat{Dec}$ to generate the final prediction $\hat{y}$:

$$\hat{y} = \widehat{Dec}(\xi, \theta_{\widehat{Dec}}). \quad (8)$$

Meanwhile, $\xi$ replaces the stochastic noise $e_i$ in $v_i$'s TRB to fuse with $z_i$, yielding $\hat{y}_i$ for computing $L_{TRB_i}$ and updating $Enc_i$ and $Dec_i$:

$$\hat{y}_i = Dec_i([z_i, \xi]). \quad (9)$$

As established in Proposition 5.1, the loss function for updating the components in OMF (i.e., $VAE_i$, $CAN$, and $\widehat{Dec}$) to achieve optimal MIB, $\xi$, is given by:

$$L_{OMF} = \frac{1}{N} \sum_{n=1}^{N} \mathbb{E}_{\epsilon_1} \mathbb{E}_{\epsilon_2} \left[ -\log q(y^n | \xi^n) \right]$$
$$+ \beta \left( KL \left[ p(\zeta_1^n | z_1^n) || \mathcal{N}(0, I) \right] \right.$$
$$\left. + r \, KL \left[ p(\zeta_2^n | z_2^n) || \mathcal{N}(0, I) \right] \right). \quad (10)$$

where $\beta$ is a hyper-parameter constraining redundancy between $\zeta_i$ and $z_i$, and $r$ is a dynamically adjusted weight balancing the regularization of $v_2$ relative to $v_1$. The implementation of $-\log q(y|\xi)$ is task-specific, as previously stated. $KL[p(\zeta_i|z_i)||\mathcal{N}(0, I)]$ represents the KL-divergence between $\zeta_i$ and the standard normal distribution. As shown in Proposition 5.2, $r$ is explicitly computed during training as:

$$r = 1 - tanh \left( \ln \frac{1}{N} \sum_{n=1}^{N} \mathbb{E}_{\epsilon_1} \mathbb{E}_{\epsilon_2} \left[ \frac{KL(p(\hat{y}_2^n | \xi^n, z_2^n) || p(\hat{y}^n | \xi^n))}{KL(p(\hat{y}_1^n | \xi^n, z_1^n) || p(\hat{y}^n | \xi^n))} \right] \right) \quad (11)$$

Furthermore, Proposition 5.7 provides a theoretical upper bound for setting $\beta$, ensuring that our methodology achieves optimal MIB. The algorithmic workflow of the main training procedure is detailed in Appendix L.

**Inference.** During inference, the TRBs are disabled, and the trained modality-specific encoder ($Enc_i$) and OMF generate optimal MIBs for test data to be used in downstream tasks.

# 5. Theoretical Foundation

Due to space constraints, we focus on the theoretical analysis of two data modalities in this section and defer the analysis of multiple data modalities ($\geq 3$) to Appendix D.2.

## 5.1. Optimal Information Bottleneck for Multimodal Data with Imbalanced Task-Relevant Information

As proposed in (Alemi et al., 2017; Federici et al., 2020; Mai et al., 2023; Wang et al., 2019), the Information Bottleneck (IB) principle aims to optimize two key objectives:

$$(1)\ maximize\ I(y;z)\ \text{and}\ (2)\ minimize\ I(v;z) \quad (12)$$

where $y$ represents task-relevant labels, $v$ the input data, $z$ the IB encoding. The first objective maximizes $z$'s expressiveness for $y$, while the second objective enforces $z$'s conciseness. These objectives can be formulated as:

$$\max_z I(y;z)\ s.t.\ I(v;z) \leq I_c, \quad (13)$$

where $I_c$ is the information constraint that limits the amount of retained input information. Introducing a Langrange multiplier $\beta > 0$, the objective function is reformulated as:

$$\max_z I(y;z) - \beta I(v;z). \quad (14)$$

For two data modalities, we propose a modified objective function to account for imbalanced task-relevant information across modalities:

$$\min_\xi \ell(\xi) = \min_\xi -I(\xi;y) + \beta(I(\xi;v_1) + rI(\xi;v_2)),\ (15)$$

where $r > 0$ is a dynamically adjusted parameter controlling the relative regularization of $v_2$ with respect to $v_1$. In Equation (15), $v_i$ can be replaced with $z_i$. To see this point, let $\bar{v}_1$ denote the information in $v_1$ that is not encoded in $z_1$. Then, we have:

$$I(\xi;v_1) = I(\xi;z_1,\bar{v}_1) = I(\xi;z_1) + \underbrace{I(\xi;\bar{v}_1|z_1)}_{=0\ \because\ F(\xi)\ \cap F(\bar{v}_1)\ =\emptyset}$$
$$= I(\xi;z_1). \quad (16)$$

Similarly, $I(\xi;v_2) = I(\xi;z_2)$. Thus, the objective function in Equation (15) can be rewritten as:

$$\min_\xi \ell(\xi) = \min_\xi -I(\xi;y) + \beta(I(\xi;z_1) + rI(\xi;z_2)). \quad (17)$$

**Proposition 5.1** (**Variational upper bound of OMIB's objective function**). *The loss function $L_{OMF}$ in Equation (10) provides a variational upper bound for optimizing the objective function in Equation (17) and can be explicitly calculated during training.*

*Proof.* See Appendix B. ∎

Moreover, when a substantial portion of task-relevant information remains in $v_2$ relative to $v_1$, the value of $r$ should be small to encourage incorporating more information from $v_2$ in subsequent training iterations. Simultaneously, $r$ should be bounded to prevent over-regularizing information from $v_2$. Thus, $r$ can be mathematically expressed as:

$$r \propto \frac{I(y;v_1|\xi)}{I(y;v_2|\xi)}, r \in (0,u), \quad (18)$$

where $I(y;v_i|\xi)$ represents the amount of task-relevant information in $v_i$ not encoded in $\xi$, and $u$ is an upper bound. In this study, $u$ is set to 2, as it is implemented using a $tahn$ function as in Equation (11), which is justified by the following proposition.

**Proposition 5.2** (**Explicit formula for $r$**). *Equation (11) satisfies Equation (18), providing an explicit formula for computing $r$ during training.*

*Proof.* See Appendix B. ∎

In the next section, we establish a theoretical bound for $\beta$, ensuring that $\xi$ attains optimality during the optimization of the objective function in Equation (17).

## 5.2. Achievability of Optimal Multimodal Information Bottleneck

**Assumption 5.3.** As illustrated in Figure 1, given two modalities, $v_1$ and $v_2$, the task-relevant information set $\{a\}$ consists of three components: $a_0, a_1$, and $a_2$. Specifically, $a_0$ is shared between both modalities, while $a_1$ and $a_2$ are specific to $v_1$ and $v_2$, respectively. The task-relevant labels $y$ are determined by $\{a\}$. Moreover, $v_1$ and $v_2$ contain modality-specific superfluous information $b_1$ and $b_2$, respectively, in addition to shared superfluous information $b_0$.

**Definition 5.4** (**Optimal multimodal information bottleneck**). Under Assumption 5.3, the optimal MIB, $\xi_{opt}$, for $v_1$ and $v_2$ satisfies:

$$F(\xi_{opt}) = \{a_0, a_1, a_2\}, \quad (19)$$

ensuring that $\xi_{opt}$ encompasses all task-relevant information ($a_0, a_1$, and $a_2$) while exempting from superfluous information ($b_0, b_1$, and $b_2$).

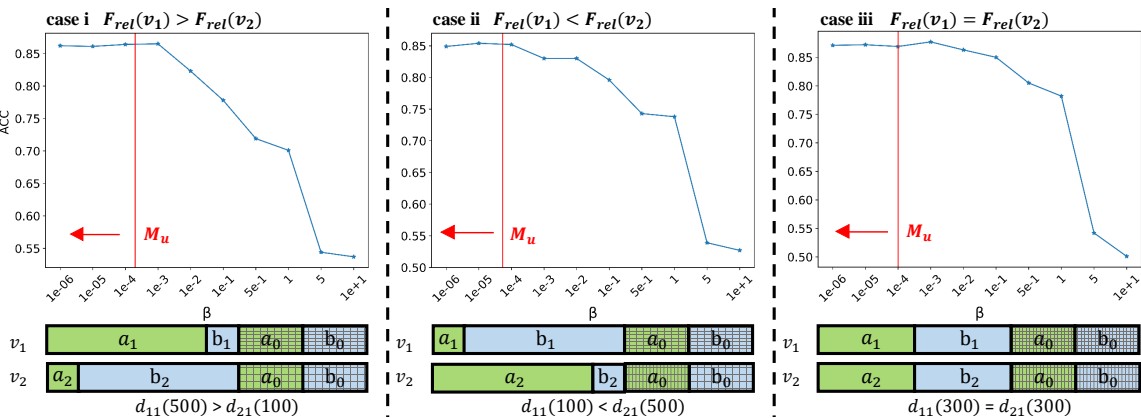

Figure 3: The impact of $\beta$ values on classification accuracy on synthetic data. $v_1$ and $v_2$ represent sample vectors of two modalities, respectively. $F_{rel}(\cdot)$ denotes task-relevant information. "$a$" sub-vectors denote task-relevant information, while "$b$" superfluous information. $d_{11}$ and $d_{21}$ denote the dimensions of modality-specific $a_1$ and $a_2$. $M_u$ is the computed $\beta$ upper bound.

.

**Lemma 5.5 (Inclusiveness of task-relevant information).** *Under Assumption 5.3, the objective function in Equation* (17) *guarantees:*

$$F(\xi) \supseteq \{a_0, a_1, a_2\}, \tag{20}$$

*provided that* $\beta \in (0, M_u]$, *where* $M_u := \frac{1}{(1+r)(H(v_1)+H(v_2)-I(v_1;v_2))}$.

*Proof.* See Appendix C $\qquad\square$

Note that $H(v_1) + H(v_2) - I(v_1; v_2)$ represents the total information encompassed by the two data modalities. Intuitively, a larger total information content requires incorporating more information from each modality into the MIB. This is achieved by setting a lower $M_u$, ensuring that all task-relevant information is included in the MIB.

**Lemma 5.6 (Exclusiveness of superfluous information).** *Under Assumption 5.3, the objective function in Equation* (17) *ensures:*

$$F(\xi) \subseteq \{a_0, a_1, a_2\} \tag{21}$$

*Proof.* See Appendix C $\qquad\square$

**Proposition 5.7 (Achievability of optimal MIB).** *Under Assumption 5.3, the optimal MIB $\xi_{opt}$ is achievable through optimization of Equation* (17) *with $\beta \in (0, M_u]$.*

*Proof.* Lemma 5.5 and Lemma 5.6 jointly demonstrate that $F(\xi) \supseteq \{a_0, a_1, a_2\}$ and $F(\xi) \subseteq \{a_0, a_1, a_2\}$, given $\beta \in (0, M_u]$. Thus, $F(\xi) = \{a_0, a_1, a_2\}$, which corresponds to $\xi_{opt}$ in Definition 5.4. This completes the proof. $\qquad\square$

In this study, we set $M_u := \frac{1}{3(H(v_1)+H(v_2)-I(v_1;v_2))} < \frac{1}{(1+r)(H(v_1)+H(v_2)-I(v_1;v_2))}$ as a tighter upper bound for $\beta$ given that $r \in (0,2)$, and $M_l := \frac{1}{3(H(v_1)+H(v_2))} \le M_u$ as a lower bound for $\beta$ to accelerate training. Importantly, both $M_l$ and $M_u$ can be computed a priori from the training data using the Mutual Information Neural Estimator (MINE, (Belghazi et al., 2018)) to estimate $H(\cdot)$ and $I(\cdot;\cdot)$ (see Appendix E).

## 6. Experiment

Due to space constraints, we defer detailed task-specific experimental settings to Appendix I and implementations of network architectures to Appendix H. Detailed descriptions of the benchmark methods and evaluation metrics are provided in Appendix J and Appendix K respectively. The best and second-best performing methods in each experiment are bolded and underlined, respectively.

Table 2: Classification accuracy of synthetic features vs. OMIB-generated MIB on simulated datasets.

| Datasets | Imbalanced (SIM-I) | balanced(SIM-III) |
|---|---|---|
| Consistent& relevant | 0.707 | 0.686 |
| Modality-specific& relevant | 0.737 | 0.744 |
| Unimodal | 0.748 / 0.82 | 0.792 / 0.78 |
| Authentic optimal MIB | **0.909** | **0.908** |
| Union of two modalities | 0.858 | 0.866 |
| OMIB-generated MIB | 0.892 | 0.890 |

### 6.1. Datasets

To facilitate the analysis of OMIB's performance and validate Proposition 5.7, we simulate three Gaussian-based two-modality dataset, **SIM-{I-III}**, for classifica-

Table 3: Comparison of multimodal fusion methods for emotion recognition on the CREMA-D.

| Methods | non-MIB-based | | | MIB-based | | | | | | OMIB |
|---|---|---|---|---|---|---|---|---|---|---|
| | Concat | BiGated | MISA | deep IB | MMIB-Cui | MMIB-Zhang | E-MIB | L-MIB | C-MIB | |
| Acc | 53.2 | 58.4 | 57.7 | 54.1 | 57.3 | 56.7 | 61.4 | 58.1 | 57.0 | **63.6** |

tion (see Appendix F). Each dataset contains all four types of information ({consistent, modality-specific} $\times$ {task-relevant, superfluous}). Moreover, they are synthesized with varying distributions of task-relevant information across modalities.

The emotion recognition experiment is conducted on **CREMA-D** (Cao et al., 2014), an audio-visual dataset in which actors express six basic emotions—happy, sad, anger, fear, disgust, and neutral—through both facial expressions and speech. The MSA experiment utilizes **CMU-MOSI** (Zadeh et al., 2016), which encompasses visual, acoustic, and textual modalities, with sentiment intensity annotated on a scale from -3 to 3. The pathological tissue detection experiment involves eight datasets derived from healthy human breast tissues (**10x-hNB-{A-H}**) and human breast cancer tissues (**10x-hBC-{A-H}**) (Xu et al., 2024b), where each dataset comprises gene expression and histology modalities. OMIB is trained on the healthy datasets and applied to the cancer datasets for pathological tissue detection. Detailed descriptions of these datasets are provided in Appendix G and Table 7.

Table 4: Comparison of multimodal fusion methods for sentiment analysis on the CMU-MOSI dataset.

| Method | Acc7 ($\uparrow$) | Acc2 ($\uparrow$) | F1($\uparrow$) | MAE($\downarrow$) | Corr($\uparrow$) |
|---|---|---|---|---|---|
| Concat | 41.5 | 81.1 | 82.0 | 0.797 | 0.745 |
| BiGated | 41.8 | 82.1 | 83.2 | 0.787 | 0.738 |
| MISA | 42.3 | 83.4 | 83.6 | 0.783 | 0.761 |
| deep IB | 45.3 | 83.2 | 83.3 | 0.747 | 0.785 |
| MMIB-Cui | 45.7 | 84.3 | 84.4 | 0.726 | 0.782 |
| MMIB-Zhang | 46.3 | 85.0 | 85.0 | 0.713 | 0.788 |
| DMIB | 40.4 | 83.2 | 83.3 | 0.810 | 0.784 |
| E-MIB | **48.6** | 85.3 | 85.3 | 0.711 | 0.798 |
| L-MIB | 45.8 | 84.6 | 84.6 | 0.732 | 0.790 |
| C-MIB | 48.2 | 85.2 | 85.2 | 0.728 | 0.793 |
| OMIB | **48.6** | **86.9** | **87.1** | **0.709** | **0.802** |

### 6.2. Empirical Analysis of OMIB Performance Using Synthetic Data

To empirically validate the effectiveness of our proposed $\beta$'s upper bound in achieving optimal MIB, we simulate three two-modality datasets (**SIM-{I-III}**) corresponding to three experimental cases (**case i-iii**) (see Appendix F). Regarding task-relevant information, Modality I dominates Modality II in SIM-I, Modality II dominates Modality I in SIM-II, and both modalities contribute equally in SIM-III, thereby covering the three primary cross-modal task-relevant information

distributions observed in practice. Each dataset is designed for a binary classification task with labels $y \in \{0, 1\}$. In each experimental case, $\beta$ is gradually increased from $10^{-6}$ to 10, well exceeding the proposed upper bound $M_u$. The generated MIBs are fed into the trained OMF prediction head to predict $y$ during testing. As shown in Figure 3, the prediction accuracy consistently peaks across all cases when using MIBs generated with $\beta$ near or below $M_u$, but rapidly declines as $\beta$ further increases. This observation aligns with our theoretical analysis, empirically confirming that optimal MIB is achievable when $\beta \leq M_u$. Notably, since $M_u$ is a tight upper bound, peak performance may still be observed for $\beta$ values slightly above $M_u$.

As detailed in Appendix F, let $x_1 = [a_0; b_0; a_1; b_1]$ and $x_2 = [a_0; b_0; a_2; b_2]$ denote feature vectors of two observations in Modality I and II, respectively. Here, $a_0$ and $b_0$ correspond to the task-relevant and superfluous sub-vectors shared by both modalities. $a_1, a_2$ are modality-specific, task-relevant sub-vectors, while $b_1, b_2$ are modality-specific, superfluous sub-vectors. By design, the authentic optimal MIB is $[a_0; a_1; a_2]$, which is used to predict $y$ and compared against the prediction using OMIB-generated MIB. Additionally, we evaluate prediction accuracy using other feature sub-vectors, including unimodal information ($x_1$ or $x_2$), consistent task-relevant information ($[a_0]$), modal-specific task-relevant information ($[a_1; a_2]$), and complete information ($[a_0; b_0; a_1; b_1; a_2; b_2]$). This experiment is conducted using SIM-I and SIM-II, corresponding to the cases of imbalanced and balanced task-relevant information, respectively. Table 2 demonstrates that OMIB-generated MIB achieves prediction accuracy most comparable to the authentic optimal MIB, surpassing all other feature sub-vector configurations that either omit task-relevant information or include superfluous information. These results further validate the optimality of OMIB-generated MIB.

### 6.3. Emotion Recognition

Here, we compare the accuracy of classifying actors' emotion types in the CREMA-D dataset using OMIB and ten benchmark methods, including three non-MIB-based fusion methods (concatenation, FiLM (Perez et al., 2018), and BiGated (Kiela et al., 2018)) and seven MIB-based state-of-the-art (SOTA) methods (E-MIB, L-MIB, and C-MIB (Mai et al., 2023) ). The classification accuracy of each method is reported in Table 3. OMIB outperforms all other methods, achieving improvements of 8.9% and 3.6% over the best-performing non-MIB-based (concatenation) and MIB-based

Table 5: Comparison of multimodal fusion methods for anomalous tissue detection performance on the 10x-hBC-{A-D} datasets

| Target Dataset | Metric | non-MIB-based | | | MIB-based | | | | | | | OMIB |
|---|---|---|---|---|---|---|---|---|---|---|---|---|
| | | Concat | BiGated | MISA | deep IB | MMIB-Cui | MMIB-Zhang | DMIB | E-MIB | L-MIB | C-MIB | |
| 10x-hBC-A | AUC | 0.537 | 0.489 | 0.498 | 0.522 | 0.623 | 0.626 | 0.423 | 0.511 | 0.598 | 0.496 | **0.728** |
| | F1 | 0.884 | 0.821 | 0.873 | 0.878 | 0.894 | 0.897 | 0.865 | 0.877 | 0.891 | 0.881 | **0.904** |
| 10x-hBC-B | AUC | 0.866 | 0.518 | 0.499 | 0.379 | 0.818 | 0.817 | 0.849 | 0.643 | 0.770 | 0.481 | **0.903** |
| | F1 | 0.654 | 0.352 | 0.213 | 0.102 | 0.559 | 0.583 | 0.607 | 0.330 | 0.483 | 0.213 | **0.663** |
| 10x-hBC-C | AUC | 0.638 | 0.563 | 0.586 | 0.433 | **0.765** | 0.662 | 0.743 | 0.598 | 0.659 | 0.511 | 0.743 |
| | F1 | 0.750 | 0.727 | 0.754 | 0.693 | 0.822 | 0.783 | **0.827** | 0.759 | 0.786 | 0.723 | 0.820 |
| 10x-hBC-D | AUC | 0.555 | 0.540 | 0.495 | 0.484 | 0.501 | 0.604 | 0.642 | 0.530 | **0.652** | 0.503 | 0.640 |
| | F1 | 0.509 | 0.494 | 0.450 | 0.443 | 0.465 | 0.524 | 0.540 | 0.483 | **0.564** | 0.465 | 0.561 |
| Mean | AUC | 0.649 | 0.528 | 0.520 | 0.455 | 0.677 | 0.677 | 0.664 | 0.571 | 0.602 | 0.498 | **0.754** |
| | F1 | 0.699 | 0.599 | 0.573 | 0.529 | 0.685 | 0.697 | 0.710 | 0.612 | 0.681 | 0.571 | **0.737** |

(E-MIB) fusion methods, respectively. These results underscore OMIB's superiority in enhancing emotion recognition performance.

### 6.4. Multimodal Sentiment Analysis

To evaluate OMIB's effectiveness in improving downstream tasks involving three modalities, we conduct MSA on the CMU-MOSI dataset, which includes visual, acoustic, and textual modalities. Specifically, OMIB and the same benchmark methods from Section 6.3 are used to predict a real-valued sentiment intensity score for each utterance, ranging from -3 to 3. Evaluation metrics for this experiment are mentioned in Appendix K. Additionally, OMIB consistently outperforms all benchmark methods across all evaluation metrics, highlighting its ability to generate high-quality MIB in a three-modal setting for enhanced regression tasks such as the MSA.

### 6.5. Anomalous Tissue Detection

In this experiment, we aim to identify anomalous tissue regions from the eight human breast cancer datasets (10x-hBC-{A-D}), which include gene expression and histology modalities. Due to the scarcity of tissue region annotations, we adopt the SVDD strategy (Ruff et al., 2018) for anomaly detection. Specifically, the model is trained exclusively on the eight healthy datasets (10x-hNB-{A-H}) to learn a compact hypersphere in the latent space, confining the multimodal representations of inliers. The trained model is then applied to the four breast cancer target datasets, generating multimodal representations whose distances to the center of the hypersphere serve as anomaly scores, based on which anomalous regions are identified. The benchmark methods are the same as those in Section 6.3 and modified to accommodate the SVDD strategy. The implementation details of OMIB for this task, are provided in Appendix H. The detection results are evaluated using the AUC and F1 scores, calculated based on the anomalous scores (see Ap-

pendix K). Table 5 demonstrates that OMIB consistently surpasses the best-performing benchmark method by an average leap of 11.4% in AUC and 3.8% in F1-score across the target datasets, confirming its superiority in anomaly detection in a multi-modal setting.

Table 6: Ablation studies on the CREMA-D dataset.

| | w/o Warm-up | w/o cross-attn | w/o OMF | w/o $r$ | Full |
|---|---|---|---|---|---|
| Acc | 60.3 | 61.5 | 59.5 | 62.2 | **63.6** |

### 6.6. Ablation Study

To gain deeper insight into the key components of OMIB, we conduct a series of ablation experiments on the CREMA-D dataset (Table 6). First, we examine the effect of removing the warm-up training ('w/o warm-up'), which leads to a 5.5% decline in accuracy. Next, we replace the CAN with simple concatenation fusion ('w/o cross-attn'), resulting in a 3.4% drop in accuracy. We also evaluate the effect of replacing the entire OMF block with simple concatenation fusion ('w/o OMF'), which significantly degrades model performance by 6.9% in accuracy. Finally, we assign equal regularization weights to $I(\xi; z_1)$ and $I(\xi; z_2)$ by omitting $r$ ('w/o $r$') and observe a performance decline of 2.3% in accuracy. In a nutshell, the degraded performance observed after removing OMIB's key components highlights their critical roles in ensuring model performance.

### 6.7. Complexity Analysis

We first provide a theoretical analysis of OMIB's complexity. OMIB's modality-specific encoder ($Enc$), task-relevant prediction head ($Dec$ and $\widehat{Dec}$), and VAEs are implemented as Multilayer Perceptron (MLP), convolutional network, or graph convolutional network, each with a complexity of $\mathcal{O}(N)$, where $N$ denotes the number of samples (He & Sun, 2015; LeCun et al., 2002; Wu et al., 2020). For the CAN network, our implementation (see Appendix H) has a time

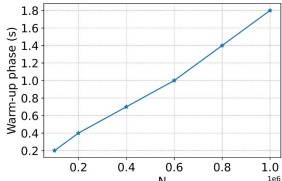 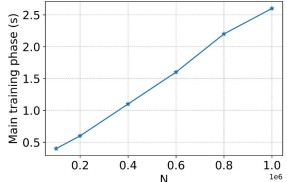

Figure 4: Runtime per epoch during warm-up and main training phase on synthetic data.

complexity of $\mathcal{O}(N \cdot M^2)$ (Vaswani et al., 2017), where $M$ represents the number of modalities. Since $M$ is typically small, $M^2$ can be treated as a constant. Thus, OMIB's overall theoretical complexity is $\mathcal{O}(N)$. We also empirically evaluate OMIB's scalability to input size using the SIM-III dataset. Explicitly, we sample six datasets with sizes: $1 \times 10^5$, $2 \times 10^5$, $4 \times 10^5$, $6 \times 10^5$, $8 \times 10^5$, and $1 \times 10^6$, while keeping the experimental settings identical to those of **case iii** in Section 6.2. We conduct separate analyses for the warm-up and main training phases, both of which demonstrate scalability to input size, as shown in Figure 4.

## 7. Conclusion

We have proposed the OMIB framework, designed to learn optimal MIB representations that effectively capture all task-relevant information. Through theoretical analysis, we demonstrate that adjusting the weights of the IB loss across different modalities facilitates the achievement of optimal MIB. Our experimental results show that OMIB outperforms existing MIB-based methods. Furthermore, our approach is robust, successfully achieving optimal MIB regardless of whether the SNRs between modalities are balanced or imbalanced.

## Impact Statement

This paper presents work whose goal is to advance the field of Machine Learning. There are many potential societal consequences of our work, none which we feel must be specifically highlighted here.

## Acknowledgments

We would like to thank Wenlin Li, Yan Lu, Zhengke Duan, and Junqi Li for their help with the experiments.

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

# Appendix

## A. Proofs of Mutual Information Properties

**Properties A.1.** *Properties of mutual information and entropy:*

***i***) $I(x; y) \geq 0, I(x; y|z) \geq 0$.

***ii***) $I(x; y, z) = I(x; y) + I(x; z|y)$.

***iii***) $I(x_1; x_2; \cdots; x_{n+1}) = I(x_1; \cdots; x_n)$
$\qquad\qquad\qquad - I(x_1; \cdots; x_n|x_{n+1})$.

***iv***) *If* $F(x_1) \cap F(x_2) = \emptyset \longrightarrow I(x_1; x_3|x_2) = I(x_1; x_3)$

***v***) *If* $F(v_2) \subseteq F(v_1) \longrightarrow I(v_1; v_2) = H(v_2)$,
$\qquad H(v_1, v_2) = F(v_1) \cup F(v_2) = F(v_1) = H(v_1)$

***vi***) *If* $H(v_2) \cap H(v_1) \neq \emptyset \longrightarrow H(v_1, v_2) = H(v_1) + H(v_2)$
$\qquad - I(v_1; v_2)$

***vii***) *If* $H(v_2) \cap H(v_1) = \emptyset \longrightarrow H(v_1, v_2) = H(v_1) + H(v_2)$
$\qquad = F(v_1) \cup F(v_2)$

*Proof.* The proofs of properties **i, ii,** and **iii** can be found in (Cover, 1999). For property **iv**, we first observe that:

$$F(y) \cap F(z) = \emptyset \longrightarrow p(y, z) = p(y)p(z) \tag{22}$$

This implies that $y$ and $z$ are statistically independent. Consequently, we have

$$
\begin{aligned}
I(y; z) &= \sum_{y,z} p(y, z) log \frac{p(y, z)}{p(y)p(z)} \\
&= \sum_{y,z} p(y, z) log \frac{p(y)p(z)}{p(y)p(z)} \\
&= \sum_{y,z} p(y, z) log\, 1 \\
&= 0
\end{aligned}
\tag{23}
$$

Given that $I(y; z) = I(x; y; z) + I(x; z|y)$, and noting that $I(x; y; z) \geq 0$ and $I(x; z|y) \geq 0$, it follows that:

$$I(y; z) = 0 \longrightarrow I(x; y; z) = 0 \text{ and } I(x; z|y) = 0 \tag{24}$$

Therefore, we obtain that:

$$I(x; y|z) = I(x; y) - \overbrace{I(x; y; z)}^{=0} = I(x; y) \tag{25}$$

For property **v**:

$$H(v_1; v_2) = \iint\limits_{v_1, v_2} p(v_1, v_2) log(\frac{p(v_1, v_2)}{p(v_1)p(v_2)})$$

$$= \iint\limits_{v_1, v_2} p(v_1, v_2) log(\frac{\overbrace{p(v_2|v_1)}^{=1 \text{ as } F(v_2) \subseteq F(v_1)} p(v_1)}{p(v_1)p(v_2)}) \tag{26}$$

$$= \int\limits_{v_2} - p(v_2) log(p(v_2)) = H(v_2).$$

In addition, for $I(v_1, v_2)$, we have:

$$H(v_1, v_2) = F(v_1) \cup F(v_2)$$

$$= \iint\limits_{v_1, v_2} - p(v_1, v_2) log(p(v_1, v_2))$$

$$= \iint\limits_{v_1, v_2} - p(v_1, v_2) log(p(v_2|v_1)p(v_1)) \tag{27}$$

$$= \int\limits_{v_1} - p(v_1) log(p(v_1)) = H(v_1) = F(v_1).$$

For property **vi**, we have:

$$H(v_1) \cap H(v_2) \neq \emptyset \longrightarrow p(v_1, v_2) \neq p(v_1)p(v_2). \tag{28}$$

The mutual information $I(v_1; v_2)$ is defined as:

$$I(v_1; v_2) = \iint p(v_1, v_2) \log \frac{p(v_1, v_2)}{p(v_1)p(v_2)} \, dv_1 \, dv_2 \neq 0$$

$$= \iint p(v_1, v_2) \log p(v_1, v_2) \, dv_1 \, dv_2 - \iint p(v_1, v_2) \log p(v_1) \, dv_1 \, dv_2$$

$$- \iint p(v_1, v_2) \log p(v_2) \, dv_1 \, dv_2$$

$$= - \iint p(v_1, v_2) \log p(v_1) \, dv_1 \, dv_2 - \iint p(v_1, v_2) \log p(v_2) \, dv_1 \, dv_2$$

$$+ \iint p(v_1, v_2) \log p(v_1, v_2) \, dv_1 \, dv_2 \tag{29}$$

$$= - \int p(v_1) \log p(v_1) \, dv_1 - \int p(v_2) \log p(v_2) \, dv_2$$

$$+ \iint p(v_1, v_2) \log p(v_1, v_2) \, dv_1 \, dv_2$$

$$= H(v_1) + H(v_2) - H(v_1, v_2)$$

Since $H(v_1) \cap H(v_2) \neq \emptyset \longrightarrow I(v_1; v_2) \neq 0$, we have $H(v_1, v_2) = H(v_1) + H(v_2) - I(v_1; v_2)$.

For property **vii**, we first clarified that:

$$F(v_2) \cap F(v_1) = \emptyset \longrightarrow p(v_1, v_2) = p(v_1)p(v_2) \tag{30}$$

Therefore, we have:

$$
\begin{aligned}
H(v_1, v_2) &= \iint_{v_1, v_2} - p(v_1, v_2) log(p(v_1, v_2)) \\
&= \iint_{v_1, v_2} - p(v_1)p(v_2) log(p(v_1)p(v_2)) \\
&= \int_{v_1} - p(v_1) log(p(v_1)) + \int_{v_2} - p(v_2) log(p(v_2)) \\
&= H(v_1) + H(v_2)
\end{aligned}
\tag{31}
$$

$\square$

## B. Proofs of Proposition 5.1 and Proposition 5.2

For convenient reading, the equations used in the proofs are copied from the main text:

$$
L_{OMF} = \frac{1}{N} \sum_{n=1}^{N} \mathbb{E}_{\epsilon_1} \mathbb{E}_{\epsilon_2} \left[ -\log q(y^n|\xi^n) \right] + \beta \left( KL \left[ p(\zeta_1^n|z_1^n) || \mathcal{N}(0, I) \right] + r \, KL \left[ p(\zeta_2^n|z_2^n) || \mathcal{N}(0, I) \right] \right).
\tag{32}
$$

which is copied from Equation (10).

$$
r = 1 - tanh \left( \ln \frac{1}{N} \sum_{n=1}^{N} \mathbb{E}_{\epsilon_1} \mathbb{E}_{\epsilon_2} \left[ \frac{KL(p(\hat{y}_2^n|\xi^n, z_2^n) || p(\hat{y}^n|\xi^n))}{KL(p(\hat{y}_1^n|\xi^n, z_1^n) || p(\hat{y}^n|\xi^n))} \right] \right),
\tag{33}
$$

which is copied from Equation (11).

$$
\min_\xi \ell(\xi) = \min_\xi -I(\xi; y) + \beta(I(\xi; z_1) + rI(\xi; z_2)),
\tag{34}
$$

which is copied from Equation (17)

$$
r \propto \frac{I(y; v_1|\xi)}{I(y; v_2|\xi)},
\tag{35}
$$

which is copied from Equation (18).

**Proposition B.1** (**Proposition 5.1 restated**). *The loss function, $L_{OMF}$, in Equation (32) provides a variational upper bound for optimizing the objective function in Equation (34), and can be explicitly calculated during training.*

*Proof.* For $I(\xi; y)$, we have:

$$
\begin{aligned}
I(\xi; y) &= \int dy d\xi p(y, \xi) log \frac{p(y, \xi)}{p(y)p(\xi)} \\
&= \int dy d\xi p(y, \xi) log \frac{p(y|\xi)}{p(y)}
\end{aligned}
\tag{36}
$$

Let $q(y|\xi)$ be a variational approximation to $p(y|\xi)$, and we have:

$$
KL[p(y|\xi) || q(y|\xi)] \geq 0 \Rightarrow \int dy p(y|\xi) \log p(y|\xi) \geq \int dy p(y|\xi) \log q(y|\xi)
\tag{37}
$$

Based on the above inequality, we have (Alemi et al., 2017):

$$
\begin{aligned}
I(\xi; y) &\geq \int dy d\xi p(y, \xi) log \frac{q(y|\xi)}{p(y)} \\
&= \int dy d\xi p(y, \xi) log \, q(y|\xi) - \int dy d\xi p(y, \xi) log \, p(y) \\
&= \int dy d\xi p(y, \xi) log \, q(y|\xi) + H(Y)
\end{aligned}
\tag{38}
$$

$H(Y)$ can be ignored as it is fixed during training. Therefore:

$$
\begin{aligned}
I(\xi; y) &\geq \int dy d\xi p(y, \xi) log\, q(y|\xi) \\
&= \int dy d\xi d\zeta_1 d\zeta_2 dz_1 dz_2 p(z_1, z_2, y, \zeta_1, \zeta_2, \xi) log\, q(y|\xi)
\end{aligned}
\tag{39}
$$

Furthermore, because $\xi$ is a function of $\zeta_1$ and $\zeta_2$ (i.e., $\xi = CAN(\zeta_1, \zeta_2)$), we have $I(\xi; z_1) \leq I(\zeta_1, \zeta_2; z_1)$ and $I(\xi; z_2) \leq I(\zeta_1, \zeta_2; z_2)$. Using the Markov property, we have $\zeta_1 \perp z_2$ and $\zeta_2 \perp z_1$, which leads to:

$$
I(\xi; z_1) \leq I(\zeta_1, \zeta_2; z_1) = I(\zeta_1; z_1) + \overbrace{I(\zeta_2; z_1|\zeta_1)}^{\zeta_2 \perp z_1} = I(\zeta_1; z_1)
\tag{40}
$$

Similarly, $I(\xi; z_2) \leq I(\zeta_2; z_2)$. Therefore:

$$
I(\xi; z_i) \leq I(\zeta_i; z_i) = \int d\zeta_i dz_i p(\zeta_i, z_i) log\, \frac{p(\zeta_i|z_i)}{p(\zeta_i)}, \forall i \in \{1, 2\}
\tag{41}
$$

Let $r(\zeta_i) \sim \mathcal{N}(0, I)$ be a variational approximation to $p(\zeta_i)$, we have:

$$
\begin{aligned}
I(\xi; z_i) \leq I(\zeta_i; z_i) &= \int d\zeta_i dz_i p(\zeta_i, z_i) log\, p(\zeta_i|z_i) - \int p(\zeta_i) \log p(\zeta_i) d\zeta_i \\
&\leq \int d\zeta_i dz_i p(\zeta_i, z_i) log\, p(\zeta_i|z_i) - \int p(\zeta_i) \log r(\zeta_i) d\zeta_i \\
&= \int d\zeta_i dz_i p(\zeta_i, z_i) log\, \frac{p(\zeta_i|z_i)}{\mathcal{N}(0, I)}, \forall i \in \{1, 2\}.
\end{aligned}
\tag{42}
$$

Put Equation (39) and Equation (42) together, we have:

$$
\begin{aligned}
L = -\,&I(\xi; y) + \beta \Big( I(\xi; z_1) + r I(\xi; z_2) \Big) \\
\leq -&\int dy dz_1 dz_2 p(y, z_1, z_2) \int d\xi d\zeta_1 d\zeta_2 p(\xi|\zeta_1, \zeta_2) p(\zeta_1|z_1) p(\zeta_2|z_2) log\, q(y|\xi) \\
+\,&\beta \Big( \int dz_1 p(z_1) \int d\zeta_1 p(\zeta_1|z_1) log\, \frac{p(\zeta_1|z_1)}{\mathcal{N}(0, I)} + r \int dz_2 p(z_2) \int d\zeta_2 p(\zeta_2|z_2) log\, \frac{p(\zeta_2|z_2)}{\mathcal{N}(0, I)} \Big)
\end{aligned}
\tag{43}
$$

Note that $p(z_1, z_2, y), p(z_1)$, and $p(z_2)$ can be approximated using the empirical data distribution (Alemi et al., 2017; Wang et al., 2019), which leads to the objective function:

$$
\begin{aligned}
L \approx \frac{1}{N} \sum_{n=1}^{N} \Big[ &-\int d\xi d\zeta_1 d\zeta_2\, p(\xi^n|\zeta_1^n, \zeta_2^n) p(\zeta_1^n|z_1^n) p(\zeta_2^n|z_2^n) log\, q(y^n|\xi^n) \\
&+ \beta \Big( \int d\zeta_1 p(\zeta_1^n|z_1^n) log\, \frac{p(\zeta_1^n|z_1^n)}{\mathcal{N}(0, I)} + r \int d\zeta_2 p(\zeta_2^n|z_2^n) log\, \frac{p(\zeta_2^n|z_2^n)}{\mathcal{N}(0, I)} \Big) \Big]
\end{aligned}
\tag{44}
$$

Given $\zeta_i = \mu_i + \Sigma_i \times \epsilon_i$ in Equation (6), we have:

$$
\begin{aligned}
\mathcal{L} &= \frac{1}{N} \sum_{n=1}^{N} \mathbb{E}_{\epsilon_1} \mathbb{E}_{\epsilon_2} \left[ -\log q(y^n|\xi^n) \right] + \beta \Big( KL \left[ p(\zeta_1^n|z_1^n)||\mathcal{N}(0, I) \right] + r\, KL \left[ p(\zeta_2^n|z_2^n)||\mathcal{N}(0, I) \right] \Big) \\
&= L_{OMF}
\end{aligned}
\tag{45}
$$

This completes the proof. $\qquad\square$

**Proposition B.2 (Proposition 5.2 restated).** *Equation (33) satisfies Equation (35), thus providing an explicit formula for computing $r$ during training.*

*Proof.* Firstly, $z_1$ and $z_2$ are sufficient encodings of modalities $v_1$ and $v_2$ for $y$, respectively. Let $\bar{v}_i$ represent the superfluous information in $v_i$ that is not encoded in $z_i$. Then, we have:

$$
\begin{aligned}
I(y; v_i|\xi) &= I(y; z_i, \bar{v}_i|\xi) \\
&= I(y; z_i|\xi) + \underbrace{I(y; \bar{v}_i|z_i, \xi)}_{=0 \, \because \, F(y) \cap F(\bar{v}_i) = \emptyset} \\
&= I(y; z_i|\xi), \forall i \in \{1, 2\}.
\end{aligned}
\tag{46}
$$

Let $z_4 = \{z_1, \xi\}$ and $z_5 = \{z_2, \xi\}$, then we have:

$$
\begin{aligned}
I(y; v_1|\xi) &= I(y; z_1|\xi) = I(z_1, \xi; y) - I(\xi; y) = I(z_4; y) - I(\xi; y), \\
I(y; v_2|\xi) &= I(y; z_2|\xi) = I(z_2, \xi; y) - I(\xi; y) = I(z_5; y) - I(\xi; y).
\end{aligned}
\tag{47}
$$

Then $I(y; z_1|\xi)$ can be expressed as:

$$
\begin{aligned}
I(y; z_1|\xi) &= I(z_4; y) - I(\xi; y) \\
&= H(y) - H(y|z_4) - H(y) + H(y|\xi) \\
&= H(y|\xi) - H(y|z_4) \\
&= -\int p(\xi)d\xi \int p(y|\xi)\log p(y|\xi)dy + \int p(z_4)dz_4 \int p(y|z_4)\log p(y|z_4)dy \\
&= -\iint p(\xi)p(y|\xi)\log\left[p(y|z_4)\frac{p(y|\xi)}{p(y|z_4)}\right]d\xi dy \\
&\quad + \iint p(z_4)p(y|z_4)\log\left[p(y|\xi)\frac{p(y|z_4)}{p(y|\xi)}\right]dz_4 dy \\
&= -\iint p(\xi)p(y|\xi)\log\frac{p(y|\xi)}{p(y|z_4)}d\xi dy - \iint p(\xi)p(y|\xi)\log p(y|z_4)d\xi dy \\
&\quad + \iint p(z_4)p(y|z_4)\log\frac{p(y|z_4)}{p(y|\xi)}dz_4 dy + \iint p(z_4)p(y|z_4)\log p(y|\xi)dz_4 dy \\
&= -\int p(\xi)KL(p(y|\xi)||p(y|z_4))d\xi - \int p(y)\log p(y|z_4)dy \\
&\quad + \int p(z_4)KL(p(y|z_4)||p(y|\xi))dz_4 + \int p(y)\log p(y|\xi)dy \\
&= \int p(z_4)KL(p(y|z_4)||p(y|\xi))dz_4 + \int p(y)\log\frac{p(y|\xi)}{p(y|z_4)}dy \\
&\quad - \int p(\xi)KL(p(y|\xi)||p(y|z_4))d\xi \\
&= \int p(z_4)KL(p(y|z_4)||p(y|\xi))dz_4 + \int p(\xi)p(y|\xi)\log\frac{p(y|\xi)}{p(y|z_4)}dyd\xi \\
&\quad - \int p(\xi)KL(p(y|\xi)||p(y|z_4))d\xi \\
&= \int p(z_4)KL(p(y|z_4)||p(y|\xi))dz_4 + \int p(\xi)KL(p(y|\xi)||p(y|z_4))d\xi \\
&\quad - \int p(\xi)KL(p(y|\xi)||p(y|z_4))d\xi \\
&= \int p(z_4)KL(p(y|z_4)||p(y|\xi))dz_4 \\
&= \mathbb{E}_{z_4}[KL(p(y|z_4)||p(y|\xi))]
\end{aligned}
\tag{48}
$$

Similarly, $I(y; z_2|\xi) = \mathbb{E}_{z_5}[KL(p(y|z_5)||p(y|\xi))]$, and $\frac{I(y;v_1|\xi)}{I(y;v_2|\xi)}$ can be calculated as:

$$
\begin{aligned}
\frac{I(y; v_2|\xi)}{I(y; v_1|\xi)} &= \frac{\mathbb{E}_{z_5}[KL(p(y|z_5) \parallel p(y|\xi))]}{\mathbb{E}_{z_4}[KL(p(y|z_4) \parallel p(y|\xi))]} \\
&= \frac{1}{N} \sum_{n=1}^{N} \mathbb{E}_{\epsilon_1} \mathbb{E}_{\epsilon_2} \left[ \frac{KL(p(\hat{y}_2^n|\xi^n, z_2^n)||p(\hat{y}^n|\xi^n))}{KL(p(\hat{y}_1^n|\xi^n, z_1^n)||p(\hat{y}^n|\xi^n))} \right]
\end{aligned}
\tag{49}
$$

Finally, we have:

$$
\begin{aligned}
r &= 1 - tanh\left( \ln \frac{1}{N} \sum_{n=1}^{N} \mathbb{E}_{\epsilon_1} \mathbb{E}_{\epsilon_2} \left[ \frac{KL(p(\hat{y}_2^n|\xi^n, z_2^n)||p(\hat{y}^n|\xi^n))}{KL(p(\hat{y}_1^n|\xi^n, z_1^n)||p(\hat{y}^n|\xi^n))} \right] \right) \\
&= 1 - tanh(\ln \frac{I(y; v_2|\xi)}{I(y; v_1|\xi)}) \propto \frac{I(y; v_1|\xi)}{I(y; v_2|\xi)}
\end{aligned}
\tag{50}
$$

This completes the proof. $\square$

## C. Proofs of Lemma 5.5 and Lemma 5.6

As proposed in Section 5.1, the objective function of MIB can be written as:

$$
\min_{\xi} \ell(\xi) = \min_{\xi} -I(\xi; y) + \beta(I(\xi; z_1) + rI(\xi; z_2))
\tag{17}
$$

Based on Assumption 5.3 in Section 5.2, we have:

$$
\begin{aligned}
&F(y) = \{a\} = \{a_0, a_1, a_2\}, \\
&F(v_1) = \{a_0, a_1, b_1, b_0\}, F(v_2) = \{a_0, a_2, b_2, b_0\}, \\
&\{a_0\} \cap \{a_1\} = \emptyset, \{a_0\} \cap \{a_2\} = \emptyset, \{a_1\} \cap \{a_2\} = \emptyset, \\
&\{b_i\} \cap \{a_0\} = \emptyset, \{b_i\} \cap \{a_1\} = \emptyset, \{b_i\} \cap \{a_2\} = \emptyset, \forall i \in \{0, 1, 2\} \\
&I(y; v_1) = \{a\} \cap F(v_1) = \{a_0, a_1\}, \\
&I(y; v_2) = \{a\} \cap F(v_2) = \{a_0, a_2\}.
\end{aligned}
$$

**Definition C.1.** The relative mutual information between encoding $z$ and task-relevant label $y$ is defined as the ratio of their mutual information to their total information:

$$
\widehat{I}(z; y) = \frac{I(z; y)}{F(z) \cup F(y)} = \frac{I(z; y)}{F(z, y)} = \frac{I(z; y)}{H(z) + H(y) - I(z; y)}
$$

Compared to mutual information, relative mutual information more accurately reflects the amount of task-relevant information (i.e., $I(\xi; y)$) in total information (i.e., $F(\xi) \cup F(y)$), which aligns more with the objective of maximizing task-relevant information in MIB. Consequently, we replace $I(\xi; y)$ with $\widehat{I}(\xi; y)$ in Equation (17) in the following analysis.

**Lemma C.2 (Lemma 5.5 restated).** *Under Assumption 5.3, the objective function in Equation* (17) *ensures:*

$$
F(\xi) \supseteq \{a_0, a_1, a_2\},
\tag{51}
$$

*when $\beta \leq M_u$, where $M_u := \frac{1}{(1+r)(H(v_1)+H(v_2)-I(v_1;v_2))}$.*

*Proof.* Let $\{\check{\xi}_1\} = (\{a_0, a_1, a_2\}/(\{a_0, a_1, a_2\} \cap F(\xi))) \cap \{a_1\}$ represent the task-relevant information in $a_1$ that is not included in $\xi$. It is obvious:

$$
\begin{aligned}
&\{\check{\xi}_1\} \subset F(y), \{\check{\xi}_1\} \cap F(\xi) = \emptyset, \\
&\{\check{\xi}_1\} \cap \{a_0\} = \emptyset, \{\check{\xi}_1\} \cap \{a_2\} = \emptyset, \\
&\{\check{\xi}_1\} \cap F(v_2) = \emptyset, \{\check{\xi}_1\} \cap F(z_2) = \emptyset.
\end{aligned}
\tag{52}
$$

If $\{\check{\xi}_1\} \neq \emptyset$, let $\xi' = \{\xi, \check{\xi}_1\}$. Using properties in Appendix A, we have:

$$I(\xi; v_1) - I(\xi'; v_1) = I(\xi; v_1) - I(\xi, \check{\xi}_1; v_1)$$

$$= I(\xi; v_1) - I(\xi; v_1) - \overbrace{I(v_1; \check{\xi}_1 | \xi)}^{\{\check{\xi}_1\} \cap F(\xi) = \emptyset (52)}$$

$$= -I(v_1; \check{\xi}_1) < 0$$

$$I(\xi; v_2) - I(\xi'; v_2) = I(\xi; v_2) - I(\xi, \check{\xi}_1; v_2)$$

$$= I(\xi; v_2) - I(\xi; v_2) - \overbrace{I(v_2; \check{\xi}_1 \mid \xi)}^{\begin{subarray}{c}\because \{\check{\xi}_1\} \cap F(v_2) = \emptyset (52) \\ \therefore I(v_2; \check{\xi}_1 | \xi) = 0\end{subarray}}$$

$$= 0$$

$$\widehat{I}(\xi'; y) - \widehat{I}(\xi; y) = \frac{I(\xi, \check{\xi}_1; y)}{F(\xi, \check{\xi}_1, y)} - \frac{I(\xi; y)}{F(\xi, y)}$$

$$= \frac{I(\xi, \check{\xi}_1; y)}{F(\xi, \check{\xi}_1, y)} - \frac{I(\xi; y)}{\underbrace{F(\xi, y) \cup F(\check{\xi}_1)}_{=F(\xi, y) \text{ as } \{\check{\xi}_1\} \subset F(y)(52)}}$$

$$= \frac{\overbrace{I(y; \check{\xi}_1|\xi)}^{\{\check{\xi}_1\} \cap F(\xi) = \emptyset \ (52)}}{F(\xi, \check{\xi}_1, y)}$$

$$= \frac{I(y; \check{\xi}_1)}{F(\xi, \check{\xi}_1, y)} > 0$$

For $\ell(\xi) - \ell(\xi')$, we have:

$$\ell(\xi) - \ell(\xi') = \widehat{I}(\xi'; y) - \widehat{I}(\xi; y) + \beta(I(\xi; v_1) - I(\xi'; v_1) + rI(\xi; v_2) - rI(\xi'; v_2))$$

$$= \frac{I(y; \check{\xi}_1)}{F(\xi, \check{\xi}_1, y)} - \beta I(v_1; \check{\xi}_1)$$

When $\beta < \frac{I(y; \check{\xi}_1)}{I(v_1; \check{\xi}_1) F(\xi, \check{\xi}_1, y)}$, $\ell(\xi) - \ell(\xi') > 0$, so optimizing the loss function will drive $\xi$ toward $\xi'$ until $\{\check{\xi}_1\} = \emptyset$, namely $F(\xi) \supseteq \{a_1\}$. We further suppose $\{\check{\xi}_2\} = (\{a_0, a_1, a_2\}/(\{a_0, a_1, a_2\} \cap F(\xi))) \cap \{a_2\}$ represent the task-relevant information in $a_2$ that is not included in $\xi$. Similarly, if $\{\check{\xi}_2\} \neq \emptyset$ and $\beta < \frac{I(y; \check{\xi}_2)}{rI(v_2; \check{\xi}_2) F(\xi, \check{\xi}_2, y)}$, the optimization will update $\xi$ until $\{\check{\xi}_2\} = \emptyset$, namely $F(\xi) \supseteq \{a_2\}$.

Moreover, let $\{\check{\xi}_0\} = (\{a_0, a_1, a_2\}/(\{a_0, a_1, a_2\} \cap F(\xi))) \cap \{a_0\}$ represent the task-relevant information in $a_0$ that is not included in $\xi$. If $\{\check{\xi}_0\} \neq \emptyset$, let $\xi' = \{\xi, \check{\xi}_0\}$. Then we have:

$$I(\xi; v_i) - I(\xi'; v_i) = I(\xi; v_i) - I(\xi, \check{\xi}_0; v_i)$$

$$= I(\xi; v_i) - I(\xi; v_i) - \overbrace{I(v_i; \check{\xi}_0|\xi)}^{\{\check{\xi}_0\} \cap F(\xi) = \emptyset}$$

$$= -I(v_i; \check{\xi}_0) < 0, \forall i \in \{1, 2\}.$$

$$\widehat{I}(\xi'; y) - \widehat{I}(\xi; y) = \frac{I(\xi, \check{\xi}_0; y)}{F(\xi, \check{\xi}_0, y)} - \frac{I(\xi; y)}{F(\xi, y)}$$

$$= \frac{I(\xi, \check{\xi}_0; y)}{F(\xi, \check{\xi}_0, y)} - \frac{I(\xi; y)}{\underbrace{F(\xi, y) \cup F(\check{\xi}_0)}_{=F(\xi, y) \text{ as } \{\check{\xi}_0\} \subset F(y)}}$$

$$= \frac{\overbrace{I(y; \check{\xi}_0 | \xi)}^{\{\check{\xi}_0\} \cap F(\xi) = \emptyset}}{F(\xi, \check{\xi}_0, y)}$$

$$= \frac{I(y; \check{\xi}_0)}{F(\xi, \check{\xi}_0, y)} > 0$$

For $\ell(\xi) - \ell(\xi')$, we have:

$$\ell(\xi) - \ell(\xi') = \widehat{I}(\xi'; y) - \widehat{I}(\xi; y) + \beta(I(\xi; v_1) - I(\xi'; v_1) + rI(\xi; v_2) - rI(\xi'; v_2))$$

$$= \frac{I(y; \check{\xi}_0)}{F(\xi, \check{\xi}_0, y)} - \beta(I(v_1; \check{\xi}_0) + rI(v_2; \check{\xi}_0))$$

Therefore, when $\beta < \frac{I(y; \check{\xi}_0)}{F(\xi, \check{\xi}_0, y)(I(v_1; \check{\xi}_0) + rI(v_2; \check{\xi}_0))}$, the optimization will update $\xi$ until $\{\check{\xi}_0\} = \emptyset$, namely $F(\xi) \supseteq \{a_0\}$.

Put together, the optimization procedure ensures $F(\xi) \supseteq \{a_0, a_1, a_2\}$ when:

$$\beta < UB_\beta = min\Big( \frac{I(y; \check{\xi}_1)}{I(v_1; \check{\xi}_1)F(\xi, \check{\xi}_1, y)}, \frac{I(y; \check{\xi}_2)}{rI(v_2; \check{\xi}_2)F(\xi, \check{\xi}_2, y)}, \frac{I(y; \check{\xi}_0)}{F(\xi, \check{\xi}_0, y)(I(v_1; \check{\xi}_0) + rI(v_2; \check{\xi}_0))} \Big). \tag{53}$$

Finally, we prove in Lemma C.3 below that $M_u = \frac{1}{(1+r)(H(v_1)+H(v_2)-I(v_1;v_2))}$ is a lower bound of $UB_\beta$. When $\beta < M_u$, the optimization procedure guarantees $F(\xi) \supseteq \{a_0, a_1, a_2\}$. This completes the proof. $\qquad \square$

**Lemma C.3.** $UB_\beta$ in Equation (53) satisfies: $UB_\beta > M_u$, where $M_u = \frac{1}{(1+r)(H(v_1)+H(v_2)-I(v_1;v_2))}$.

$H(\cdot)$ and $I(\cdot; \cdot)$ can be estimated using MINE (Belghazi et al., 2018) (see Appendix E).

*Proof.* As shown in Equation (53)

$$UB_\beta = min\Big( \frac{I(y; \check{\xi}_1)}{I(v_1; \check{\xi}_1)F(\xi, \check{\xi}_1, y)}, \frac{I(y; \check{\xi}_2)}{rI(v_2; \check{\xi}_2)F(\xi, \check{\xi}_2, y)}, \frac{I(y; \check{\xi}_0)}{F(\xi, \check{\xi}_0, y)(I(v_1; \check{\xi}_0) + rI(v_2; \check{\xi}_0))} \Big)$$

$\because \{\check{\xi}_1\} \subseteq F(\xi, y)$, we have $F(\xi, \check{\xi}_1, y) = F(\xi, y)$ so that:

$$\frac{I(y; \check{\xi}_1)}{I(v_1; \check{\xi}_1)F(\xi, \check{\xi}_1, y)} = \frac{I(y; \check{\xi}_1)}{I(v_1; \check{\xi}_1)F(\xi, y)}$$

$\because \{\check{\xi}_1\} \subseteq \{a_1\}, \{a_1\} \subseteq \{v_1\}$, and $\{a_1\} \subseteq \{y\}, \therefore \{\check{\xi}_1\} \subseteq \{v_1\}$ and $\{\check{\xi}_1\} \subseteq \{y\}$. Then, according to property **v** in Properties A.1, $I(y; \check{\xi}_1) = H(\check{\xi}_1)$ and $I(v_1; \check{\xi}_1) = H(\check{\xi}_1)$, which leads to:

$$\frac{I(y; \check{\xi}_1)}{I(v_1; \check{\xi}_1)F(\xi, y)} = \frac{H(\check{\xi}_1)}{H(\check{\xi}_1)F(\xi, y)}$$

$$= \frac{1}{F(\xi, y)}$$

Similarly, $\frac{I(y; \check{\xi}_2)}{rI(v_2; \check{\xi}_2)F(\xi, \check{\xi}_2, y)}$ is simplify to $\frac{1}{rF(\xi, y)}$. Moreover, $F(\xi, \check{\xi}_0, y) = F(\xi, y)$ since $\{\check{\xi}_0\} \subseteq F(\xi, y)$. Then, it follows that:

$$\frac{I(y; \check{\xi}_0)}{F(\xi, \check{\xi}_0, y)(I(v_1; \check{\xi}_0) + rI(v_2; \check{\xi}_0))} = \frac{I(y; \check{\xi}_0)}{F(\xi, y)(I(v_1; \check{\xi}_0) + rI(v_2; \check{\xi}_0))}$$

$\therefore \{\check{\xi}_0\} \subseteq \{a_0\}, \{a_0\} \subseteq \{v_1\}, \{a_0\} \subseteq \{v_2\},$ and $\{a_0\} \subseteq \{y\}, \therefore \{\check{\xi}_0\} \subseteq \{v_1\}, \{\check{\xi}_0\} \subseteq \{v_2\},$ and $\{\check{\xi}_0\} \subseteq \{y\}.$ Thus, by property **v** in Properties A.1, $I(y; \check{\xi}_0) = H(\check{\xi}_0), I(v_1; \check{\xi}_0) = H(\check{\xi}_0),$ and $I(v_2; \check{\xi}_0) = H(\check{\xi}_0),$ which collectively lead to:

$$
\begin{aligned}
\frac{I(y; \check{\xi}_0)}{F(\xi, y)(I(v_1; \check{\xi}_0) + rI(v_2; \check{\xi}_0))} &= \frac{H(\check{\xi}_0)}{F(\xi, y)(H(\check{\xi}_0) + rH(\check{\xi}_0))} \\
&= \frac{1}{(1 + r)F(\xi, y)} \\
&< \min\left(\frac{1}{F(\xi, y)}, \frac{1}{rF(\xi, y)}\right), \forall r > 0 \\
&= \min\left(\frac{I(y; \check{\xi}_1)}{I(v_1; \check{\xi}_1)F(\xi, \check{\xi}_1, y)}, \frac{I(y; \check{\xi}_2)}{rI(v_2; \check{\xi}_2)F(\xi, \check{\xi}_2, y)}\right).
\end{aligned}
$$

Thus, we have:

$$
\begin{aligned}
UB_\beta &= \frac{I(y; \check{\xi}_0)}{F(\xi, y)(I(v_1; \check{\xi}_0) + rI(v_2; \check{\xi}_0))} \\
&= \frac{1}{(1 + r)F(\xi, y)} \\
&> \frac{1}{(1 + r)\underbrace{F(v_1, v_2)}_{\because F(\xi, y) \subset F(v_1, v_2)}} \\
&= \frac{1}{(1 + r)(H(v_1) + H(v_2) - I(v_1; v_2))} \\
&= M_u
\end{aligned}
$$

This completes the proof. $\qquad\square$

**Lemma C.4** (**Lemma 5.6 restated**). *Under Assumption 5.3, the objective function in Equation* (34) *is optimized when:*

$$
F(\xi) \subseteq \{a_0, a_1, a_2\} \tag{54}
$$

*Proof.* Let $\hat{z}_1$ represent superfluous information that is specific to $v_1$ and not incorporated into $\xi$. Then, we have:

$$
\begin{aligned}
&\hat{z}_1 \notin \{a_0, a_1, a_2\}, \{\hat{z}_1\} \subset F(v_1), I(\hat{z}_1; v_1) > 0, \\
&I(\hat{z}_1; y) = 0, \{\xi\} \cap \{\hat{z}_1\} = \emptyset, \{v_2\} \cap \{\hat{z}_1\} = \emptyset.
\end{aligned} \tag{55}
$$

Let $\ddot{\xi} = \{\xi, \hat{z}_1\}$. The objective function becomes:

$$
\begin{aligned}
\ell(\ddot{\xi}) &= -\widehat{I}(\ddot{\xi}; y) + \beta(I(\ddot{\xi}; v_1) + rI(\ddot{\xi}; v_2)) \\
&= -\widehat{I}(\xi, \hat{z}_1; y) + \beta(I(\xi, \hat{z}_1; v_1) + rI(\xi, \hat{z}_1; v_2))
\end{aligned} \tag{56}
$$

Then we have the following equations:

$$
\begin{aligned}
I(\xi, \hat{z}_1; v_1) - I(\xi; v_1) &= \overbrace{I(v_1; \hat{z}_1 | \xi)}^{\{\hat{z}_1\} \cap \{\xi\} = \emptyset} \\
&= I(v_1; \hat{z}_1) > 0
\end{aligned} \tag{57}
$$

$$
\begin{aligned}
I(\xi, \hat{z}_1; v_2) - I(\xi; v_2) &= \overbrace{I(v_2; \hat{z}_1 | \xi)}^{\{\hat{z}_1\} \cap \{v_2\} = \emptyset} \\
&= 0
\end{aligned} \tag{58}
$$

$$\widehat{I}(\xi;y) - \widehat{I}(\xi,\hat{z}_1;y) = \frac{I(\xi;y)}{F(\xi,y)} - \frac{I(\hat{z}_1,\xi;y)}{F(\hat{z}_1,\xi,y)}$$

$$= \frac{I(\xi;y) + \overbrace{I(y;\hat{z}_1|\xi)}^{=0 \because I(\hat{z}_1;y)=0}}{F(\xi,y)} - \frac{I(\hat{z}_1,\xi;y)}{F(\hat{z}_1,\xi,y)}$$

$$= \frac{I(\hat{z}_1,\xi;y)}{F(\xi,y)} - \frac{I(\hat{z}_1,\xi;y)}{F(\hat{z}_1,\xi,y)} \tag{59}$$

$$= \frac{I(\hat{z}_1,\xi;y)}{F(\xi,y)} - \frac{I(\hat{z}_1,\xi;y)}{\underbrace{F(\hat{z}_1) + F(\xi,y)}_{\because \hat{z}_1 \perp \{\xi,y\}}}$$

$$> 0$$

Put together, we have:

$$\ell(\ddot{\xi}) - \ell(\xi)$$
$$= (\widehat{I}(\xi;y) - \widehat{I}(\xi,\hat{z}_1;y)) + \beta\Big((\widehat{I}(\hat{z}_1,\xi;v_1) - \widehat{I}(\xi;v_1)) + r(\widehat{I}(\hat{z}_1,\xi;v_2) - \widehat{I}(\xi;v_2))\Big) \tag{60}$$
$$> 0, \text{ if } \{\hat{z}_1\} \neq \emptyset.$$

For superfluous information $\hat{z}_2$ specific to $v_2$, we arrive at the same conclusion. Finally, let $\hat{z}_0$ represent superfluous information that is shared by the two modalities and not encoded in $\xi$. Then, we have:

$$\hat{z}_0 \notin \{a_0, a_1, a_2\}, \{\hat{z}_0\} \subset F(v_1), \{\hat{z}_0\} \subset F(v_2),$$
$$I(\hat{z}_0;v_1) > 0, I(\hat{z}_0;v_2) > 0, \tag{61}$$
$$I(\hat{z}_0;y) = 0, \{\xi\} \cap \{\hat{z}_0\} = \emptyset$$

Let $\ddot{\xi} = \{\xi, \hat{z}_0\}$. The objective function becomes:

$$\ell(\ddot{\xi}) = -\widehat{I}(\ddot{\xi};y) + \beta(I(\ddot{\xi};v_1) + rI(\ddot{\xi};v_2))$$
$$= -\widehat{I}(\xi,\hat{z}_0;y) + \beta(I(\xi,\hat{z}_0;v_1) + rI(\xi,\hat{z}_0;v_2)) \tag{62}$$

Then we have the following equations:

$$\widehat{I}(\xi;y) - \widehat{I}(\xi,\hat{z}_0;y) = \frac{I(\xi;y)}{F(\xi,y)} - \frac{I(\hat{z}_0,\xi;y)}{F(\hat{z}_0,\xi,y)}$$

$$= \frac{I(\xi;y) + \overbrace{I(y;\hat{z}_0|\xi)}^{=0 \because I(\hat{z}_0;y)=0}}{F(\xi,y)} - \frac{I(\hat{z}_0,\xi;y)}{F(\hat{z}_0,\xi,y)}$$

$$= \frac{I(\hat{z}_0,\xi;y)}{F(\xi,y)} - \frac{I(\hat{z}_0,\xi;y)}{F(\hat{z}_0,\xi,y)} \tag{63}$$

$$= \frac{I(\hat{z}_0,\xi;y)}{F(\xi,y)} - \frac{I(\hat{z}_0,\xi;y)}{\underbrace{F(\hat{z}_0) + F(\xi,y)}_{\because \hat{z}_0 \perp \{\xi,y\}}}$$

$$> 0$$

$$I(\xi,\hat{z}_0;v_1) - I(\xi;v_1) = \overbrace{I(v_1;\hat{z}_0|\xi)}^{\{\hat{z}_0\}\cap\{\xi\}=\emptyset} \tag{64}$$
$$= I(v_1;\hat{z}_0) > 0$$

Similarly, we have $I(\xi, \hat{z}_0; v_2) - I(\xi; v_2) = I(v_2; \hat{z}_0) > 0$.

Put together, we have:

$$
\begin{aligned}
&\ell(\ddot{\xi}) - \ell(\xi) \\
&= (\widehat{I}(\xi; y) - \widehat{I}(\xi, \hat{z}_0; y)) + \beta\Big((\widehat{I}(\hat{z}_0, \xi; v_1) - \widehat{I}(\xi; v_1)) + r(\widehat{I}(\hat{z}_0, \xi; v_2) - \widehat{I}(\xi; v_2))\Big) \\
&> 0, \text{ if } \{\hat{z}_0\} \neq \emptyset.
\end{aligned}
\tag{65}
$$

In a nutshell, the optimization procedure continues until $\xi$ does not encompass superfluous information, shared or modality specific, from $v_1, v_2$. That is, $F(\xi) \subseteq \{a_0, a_1, a_2\}$. This completes the proof. $\qquad\square$

## D. Extension to Multiple Modalities

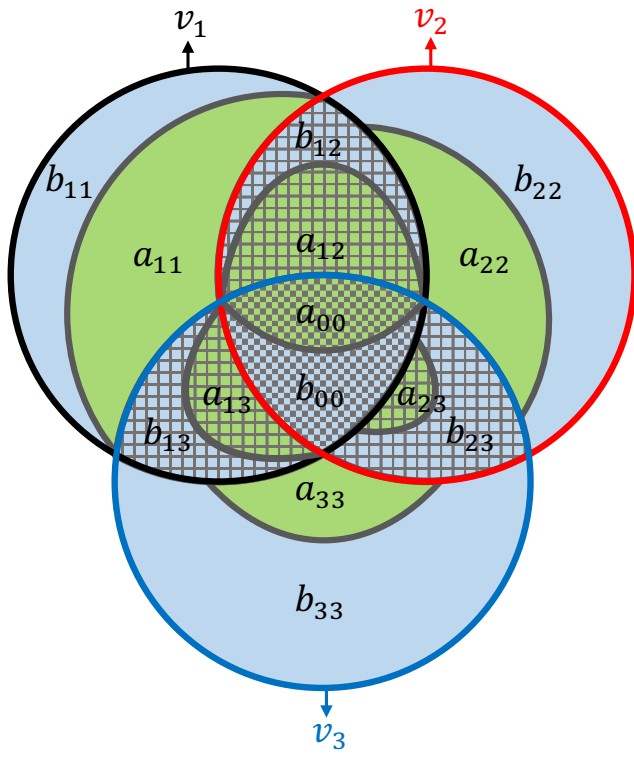

Figure 5: Venn diagrams for three data modalities ($v_1$, $v_2$, and $v_3$). The gridded area represents consistent information, while the non-gridded area denotes modality-specific information. Task-relevant information is highlighted in green, whereas superfluous information is shown in blue.

The theoretical analysis of multiple modalities ($\geq 3$) is exemplified using three modalities, $v_1$, $v_2$, and $v_3$, which yet can be readily extended to more modalities. All mathematical notations remain consistent with those in Table 1 in Section 3.

**Assumption D.1.** Given three modalities, $v_1$, $v_2$, and $v_3$, the task-relevant information set $\{a\}$ consists of seven parts—$a_{00}, a_{11}, a_{22}, a_{33}, a_{12}, a_{13}, a_{23}$, as illustrated in Figure 5. Specifically, $a_{00}$ is shared by all three modalities, while $a_{ij}$ is shared between modality pairs $(v_i, v_j)$, $\forall i, j \in \{1, 2, 3\}$, $i < j$. Meanwhile, $a_{ii}$ is specific to $v_i$, $\forall i \in \{1, 2, 3\}$. The task-relevant labels $y$ are determined by $\{a\}$. On the other hand, superfluous information is represented by $\{b\} = \{b_{00}, b_{11}, b_{22}, b_{33}, b_{12}, b_{13}, b_{23}\}$. Here, $b_{00}$ is shared by all three modalities, while $b_{ij}$ is shared between modality pairs $(v_i, v_j)$, $\forall i, j \in \{1, 2, 3\}$, $i < j$. Meanwhile, $b_{ii}$ is specific to $v_i$, $\forall i \in \{1, 2, 3\}$.

Based on the above assumption, the optimal MIB has the following definition:

**Definition D.2 (Optimal multimodal information bottleneck for three modalities).** The optimal MIB for three modalities is defined as the MIB that encompasses all task-relevant information and free of superfluous information. Let $\xi_{opt-three}$

denote the optimal MIB, and it can be explicitly expressed as:

$$F(\xi_{opt-three}) = \{a_{00}, a_{11}, a_{22}, a_{33}, a_{12}, a_{13}, a_{23}\}. \tag{66}$$

In the following sections, we first demonstrate the method for achieving the optimal MIB, followed by a theoretical analysis to establish its theoretical foundation.

### D.1. Method

The warm-up and main training phases follow those for two modalities in Section 4, except for an additional modality $v_3$. The loss function $L_{TRB}$ remains the same for each modality as in Equation (4), while the loss function $L_{OMF}$ becomes:

$$
\begin{aligned}
L_{OMF} = \frac{1}{N} \sum_{n=1}^{N} \mathbb{E}_{\epsilon_1} \mathbb{E}_{\epsilon_2} \mathbb{E}_{\epsilon_3} [-log\, q(y^n|\xi^n)] + \beta \Big( KL(p(\zeta_1^n|z_1^n) \,\|\, \mathcal{N}(0,I)) \\
+ r_1 KL(p(\zeta_2^n|z_2^n) \,\|\, \mathcal{N}(0,I)) + r_2 KL(p(\zeta_3^n|z_3^n) \,\|\, \mathcal{N}(0,I)) \Big),
\end{aligned}
\tag{67}
$$

where $\xi = CAN(\zeta_1, \zeta_2, \zeta_3, \theta_{CAN})$. Analogous to Equation (11) proved by Proposition 5.2 in Section 5.1, $r_1$ and $r_2$ are dynamic during training and explicitly calculated as:

$$
\begin{aligned}
r_1 = 1 - tanh\Big(\ln \frac{1}{N} \sum_{n=1}^{N} \mathbb{E}_{\epsilon_1} \mathbb{E}_{\epsilon_2} \Big[ \frac{KL(p(\hat{y}_2^n|\xi^n, z_2^n)\|p(\hat{y}^n|\xi^n))}{KL(p(\hat{y}_1^n|\xi^n, z_1^n)\|p(\hat{y}^n|\xi^n))} \Big] \Big), \\
r_2 = 1 - tanh\Big(\ln \frac{1}{N} \sum_{n=1}^{N} \mathbb{E}_{\epsilon_1} \mathbb{E}_{\epsilon_3} \Big[ \frac{KL(p(\hat{y}_3^n|\xi^n, z_3^n)\|p(\hat{y}^n|\xi^n))}{KL(p(\hat{y}_1^n|\xi^n, z_1^n)\|p(\hat{y}^n|\xi^n))} \Big] \Big).
\end{aligned}
\tag{68}
$$

Moreover, as proposed in Lemma D.4 in Appendix D.2, when $\beta$ in Equation (67) is upper-bounded by $M_u^2 := \frac{1}{(1+r_1+r_2)(\sum_{i=1}^{3} H(v_i) - \frac{2}{3} \sum_{1 \le i < j \le 3} I(v_i;v_j))}$, the optimization of $L_{OMF}$ ensures the achievability of $\xi_{opt}$.

### D.2. Theoretical Foundation

Under Assumption D.1, we have:

$$
\begin{aligned}
F(y) &= \{a\} = \{a_{00}, a_{11}, a_{22}, a_{33}, a_{12}, a_{13}, a_{23}\}, \\
F(v_1) &= \{a_{00}, a_{11}, a_{12}, a_{13}, b_1, b_{12}, b_{13}, b_0\}, \\
F(v_2) &= \{a_{00}, a_{22}, a_{12}, a_{23}, b_2, b_{12}, b_{23}, b_0\}, \\
F(v_3) &= \{a_{00}, a_{33}, a_{13}, a_{23}, b_3, b_{13}, b_{23}, b_0\}, \\
\{a_{ij}\} &\cap \{a_{uv}\} = \emptyset, \forall a_{ij}, a_{uv} \in \{a\}, \text{where } a_{ij} \neq a_{uv} \\
\{b_{ij}\} &\cap \{a_{uv}\} = \emptyset, \forall b_{ij} \in \{b\}, a_{uv} \in \{a\} \\
I(y;v_1) &= \{a\} \cap F(v_1) = \{a_{00}, a_{11}, a_{12}, a_{13}\}, \\
I(y;v_2) &= \{a\} \cap F(v_2) = \{a_{00}, a_{22}, a_{12}, a_{23}\}. \\
I(y;v_3) &= \{a\} \cap F(v_3) = \{a_{00}, a_{33}, a_{13}, a_{23}\}.
\end{aligned}
$$

Analogous to the analysis in Section 5.1, the objective function for obtaining optimal MIB can be formulated as:

$$\max_{\xi} \ell(\xi) = \max_{\xi} I(\xi;y) - \beta(I(\xi;v_1) + r_1 I(\xi;v_2) + r_2 I(\xi;v_3)), \tag{69}$$

which is equivalent to:

$$\min_{\xi} \ell(\xi) = \min_{\xi} -I(\xi;y) + \beta(I(\xi;z_1) + r_1 I(\xi;z_2) + r_2 I(\xi;z_3)). \tag{70}$$

The Proposition 5.1 in Section 5.1 can be trivially modified by adding the $r_2 I(\xi;z_3)$ term and applied here to establish $L_{OMF}$ in Equation (67) as a variational upper bound for Equation (70).

D.2.1. ACHIEVABILITY OF OPTIMA INFORMATION BOTTLENECK FOR THREE MODALITIES

**Lemma D.3 (Inclusiveness of task-relevant information for three modalities).** *Under Assumption D.1, the objective function in Equation* (69) *ensures:*

$$F(\xi) \supseteq \{a_{00}, a_{11}, a_{22}, a_{33}, a_{12}, a_{13}, a_{23}\}, \tag{71}$$

*when* $\beta \leq M_u^2$, *where* $M_u^2 := \frac{1}{(1+r_1+r_2)(\sum_{i=1}^{3} H(v_i) - \frac{2}{3}\sum_{1 \leq i < j \leq 3} I(v_i; v_j))}$.

*Proof.* We first analyze under which condition $\xi$ can include all the task-relevant information specific to single modality. Let $\{\check{\xi}_1\} = (\{a_{00}, a_{11}, a_{22}, a_{33}, a_{12}, a_{13}, a_{23}\}/(\{a_{00}, a_{11}, a_{22}, a_{33}, a_{12}, a_{13}, a_{23}\} \cap I(\xi))) \cap \{a_{11}\}$ represent the task-relevant information in $a_{11}$ that is not included in $\xi$. It is obvious:

$$\{\check{\xi}_1\} \subset F(y), \{\check{\xi}_1\} \cap F(\xi) = \emptyset,$$
$$\{\check{\xi}_1\} \cap \{a_{ij}\} = \emptyset, \quad \forall a_{ij} \in \{a\}/\{a_{11}\},$$
$$\{\check{\xi}_1\} \cap F(v_2) = \emptyset, \{\check{\xi}_1\} \cap F(z_2) = \emptyset,$$
$$\{\check{\xi}_1\} \cap F(v_3) = \emptyset, \{\check{\xi}_1\} \cap F(z_3) = \emptyset$$

If $\{\check{\xi}_1\} \neq \emptyset$, let $\xi' = \{\xi, \check{\xi}_1\}$ and we have:

$$\ell(\xi) = -\widehat{I}(\xi; y) + \beta(I(\xi; v_1) + r_1 I(\xi; v_2) + r_2 I(\xi; v_3))$$
$$= -\widehat{I}(\xi; y) + \beta\left(I(\xi; v_1) + r_1(I(v_2; \xi) + \overbrace{I(v_2; \check{\xi}_1|\xi)}^{=0, \because \{\check{\xi}_1\} \cap F(v_2) = \emptyset}) + r_2(I(v_3; \xi) + \overbrace{I(v_3; \check{\xi}_1|\xi)}^{=0, \because \{\check{\xi}_1\} \cap F(v_3) = \emptyset})\right)$$
$$= -\widehat{I}(\xi; y) + \beta(I(\xi; v_1) + r_1 I(\xi, \check{\xi}_1; v_2) + r_2 I(\xi, \check{\xi}_1; v_3)),$$

$$\ell(\xi) - \ell(\xi') = -\widehat{I}(\xi; y) + \beta(I(\xi; v_1) + r_1 I(\xi, \check{\xi}_1; v_2) + r_2 I(\xi, \check{\xi}_1; v_3))$$
$$- \left(-\widehat{I}(\xi, \check{\xi}_1; y) + \beta(I(\xi, \check{\xi}_1; v_1) + r_1 I(\xi, \check{\xi}_1; v_2) + r_2 I(\xi, \check{\xi}_1; v_3))\right)$$
$$= \widehat{I}(\xi, \check{\xi}_1; y) - \widehat{I}(\xi; y) + \beta(I(\xi; v_1) - I(\xi, \check{\xi}_1; v_1))$$

Using properties in Appendix A, we have:

$$\widehat{I}(\xi, \check{\xi}_1; y) - \widehat{I}(\xi; y) = \frac{I(\xi, \check{\xi}_1; y)}{F(\xi) \cup \underbrace{(F(\check{\xi}_1) \cup F(y))}_{=F(y) \text{ as } \{\check{\xi}_1\} \subset F(y)}} - \frac{I(\xi; y)}{F(\xi) \cup F(y)}$$
$$= \frac{I(\xi, \check{\xi}_1; y)}{F(\xi, y)} - \frac{I(\xi; y)}{F(\xi, y)}$$
$$= \frac{\overbrace{I(y; \check{\xi}_1|\xi)}^{\{\check{\xi}_1\} \cap \{\xi\} = \emptyset}}{F(\xi, y)}$$
$$= \frac{I(y; \check{\xi}_1)}{F(\xi, y)}$$

$$I(\xi; v_1) - I(\xi, \check{\xi}_1; v_1) = -\overbrace{I(v_1; \check{\xi}_1|\xi)}^{\{\check{\xi}_1\} \cap \{\xi\} = \emptyset} \tag{72}$$
$$= -I(v_1; \check{\xi}_1)$$

Thus:

$$
\begin{aligned}
\ell(\xi) - \ell(\xi') &= \widehat{I}(\xi, \check{\xi}_1; y) - \widehat{I}(\xi; y) + \beta I(\xi; v_1) - \beta I(\xi, \check{\xi}_1; v_1) \\
&= \frac{I(y; \check{\xi}_1)}{F(\xi, y)} - \beta I(v_1; \check{\xi}_1)
\end{aligned}
\tag{73}
$$

When $\beta < \frac{I(y;\check{\xi}_1)}{F(\xi,y)I(v_1;\check{\xi}_1)}$, $\ell(\xi) - \ell(\xi') > 0$. Therefore, optimizing the loss function will drive $\xi$ toward $\xi'$ until $\{\check{\xi}_1\} = \emptyset$, such that $F(\xi) \supseteq \{a_{11}\}$. We further suppose $\{\check{\xi}_2\} = (\{a_{00}, a_{11}, a_{22}, a_{33}, a_{12}, a_{13}, a_{23}\}/(\{a_{00}, a_{11}, a_{22}, a_{33}, a_{12}, a_{13}, a_{23}\} \cap I(\xi))) \cap \{a_{22}\}$ represent the task-relevant information in $a_{22}$ that is not included in $\xi$, and $\{\check{\xi}_3\} = (\{a_{00}, a_{11}, a_{22}, a_{33}, a_{12}, a_{13}, a_{23}\}/(\{a_{00}, a_{11}, a_{22}, a_{33}, a_{12}, a_{13}, a_{23}\} \cap I(\xi))) \cap \{a_{33}\}$ represent the task-relevant information in $a_{33}$ that is not included in $\xi$. Similarly, if $\{\check{\xi}_2\} \neq \emptyset$ and $\beta < \frac{I(y;\check{\xi}_2)}{r_1 I(v_2;\check{\xi}_2)F(\xi,\check{\xi}_2,y)}$, the optimization will update $\xi$ until $\{\check{\xi}_2\} = \emptyset$, leading to $F(\xi) \supseteq \{a_{22}\}$; and if $\{\check{\xi}_3\} \neq \emptyset$ and $\beta < \frac{I(y;\check{\xi}_3)}{r_2 I(v_3;\check{\xi}_3)F(\xi,\check{\xi}_3,y)}$, the optimization will update $\xi$ until $\{\check{\xi}_3\} = \emptyset$, leading to $F(\xi) \supseteq \{a_{33}\}$.

We then analyze under which condition $\xi$ can include all the task-relevant information shared by two modalities. Let $\{\check{\xi}_{12}\} = (\{a_{00}, a_{11}, a_{22}, a_{33}, a_{12}, a_{13}, a_{23}\}/(\{a_{00}, a_{11}, a_{22}, a_{33}, a_{12}, a_{13}, a_{23}\} \cap I(\xi))) \cap \{a_{12}\}$ represent the task-relevant information in $a_{12}$ that is not included in $\xi$. It is obvious:

$$
\begin{aligned}
&\{\check{\xi}_{12}\} \subset F(y), \{\check{\xi}_{12}\} \cap F(\xi) = \emptyset, \\
&\{\check{\xi}_{12}\} \cap \{a_{ij}\} = \emptyset, \quad \forall a_{ij} \in \{a\}/\{a_{12}\}, \\
&\{\check{\xi}_{12}\} \cap F(v_3) = \emptyset, \{\check{\xi}_{12}\} \cap F(z_3) = \emptyset
\end{aligned}
$$

If $\{\check{\xi}_{12}\} \neq \emptyset$, let $\xi' = \{\xi, \check{\xi}_{12}\}$, then we have:

$$
\begin{aligned}
\ell(\xi) &= -\widehat{I}(\xi; y) + \beta(I(\xi; v_1) + r_1 I(\xi; v_2) + r_2 I(\xi; v_3)) \\
&= -\widehat{I}(\xi; y) + \beta\Big(I(\xi; v_1) + r_1 I(v_2; \xi) + r_2(I(v_3; \xi) + \overbrace{I(v_3; \check{\xi}_{12}|\xi)}^{=0,\, \because \{\check{\xi}_{12}\} \cap F(v_3) = \emptyset})\Big) \\
&= -\widehat{I}(\xi; y) + \beta(I(\xi; v_1) + r_1 I(\xi; v_2) + r_2 I(\xi, \check{\xi}_{12}; v_3))
\end{aligned}
$$

Therefore, $\ell(\xi) - \ell(\xi')$ can be written as:

$$
\begin{aligned}
\ell(\xi) - \ell(\xi') &= -\widehat{I}(\xi; y) + \beta(I(\xi; v_1) + r_1 I(\xi; v_2) + r_2 I(\xi, \check{\xi}_{12}; v_3)) \\
&\quad - \Big(-\widehat{I}(\xi, \check{\xi}_{12}; y) + \beta(I(\xi, \check{\xi}_{12}; v_1) + r_1 I(\xi, \check{\xi}_{12}; v_2) + r_2 I(\xi, \check{\xi}_{12}; v_3))\Big) \\
&= \widehat{I}(\xi, \check{\xi}_{12}; y) - \widehat{I}(\xi; y) + \beta\Big(I(\xi; v_1) - I(\xi, \check{\xi}_{12}; v_1) + r_1(I(\xi; v_2) - I(\xi, \check{\xi}_{12}; v_2))\Big)
\end{aligned}
$$

Using properties in Appendix A, we have:

$$
\begin{aligned}
\widehat{I}(\xi, \check{\xi}_{12}; y) - \widehat{I}(\xi; y) &= \frac{I(\xi, \check{\xi}_{12}; y)}{F(\xi) \cup \underbrace{(F(y) \cup F(\check{\xi}_{12}))}_{=F(y) \text{ as } \{\check{\xi}_{12}\} \subset F(y)}} - \frac{I(\xi; y)}{F(\xi) \cup F(y)} \\
&= \frac{I(\xi, \check{\xi}_{12}; y)}{F(\xi, y)} - \frac{I(\xi; y)}{F(\xi, y)} \\
&= \frac{\overbrace{I(y; \check{\xi}_{12}|\xi)}^{\{\check{\xi}_{12}\} \cap \{\xi\} = \emptyset}}{F(\xi, y)} \\
&= \frac{I(y; \check{\xi}_{12})}{F(\xi, y)} > 0
\end{aligned}
$$

$$I(\xi; v_1) - I(\xi, \check{\xi}_{12}; v_1) = -\overbrace{I(v_1; \check{\xi}_{12}|\xi)}^{\{\check{\xi}_{12}\} \cap \{\xi\} = \emptyset} \tag{74}$$
$$= -I(v_1; \check{\xi}_{12}) < 0$$

Similarly, we obtain that $I(\xi; v_2) - I(\xi, \check{\xi}_{12}; v_2) = -I(v_2; \check{\xi}_{12})$.

Thus:

$$\ell(\xi) - \ell(\xi') = \widehat{I}(\xi, \check{\xi}_{12}; y) - \widehat{I}(\xi; y) + \beta\Big(I(\xi; v_1) - I(\xi, \check{\xi}_{12}; v_1) + r_1(I(\xi; v_2) - I(\xi, \check{\xi}_{12}; v_2))\Big)$$
$$= \frac{I(y; \check{\xi}_{12})}{F(\xi, y)} - \beta(I(v_1; \check{\xi}_{12}) + r_1 I(v_2; \check{\xi}_{12})) \tag{75}$$

When $\beta < \frac{I(y; \check{\xi}_{12})}{F(\xi,y)(I(v_1; \check{\xi}_{12}) + r_1 I(v_2; \check{\xi}_{12}))}$, $\ell(\xi) - \ell(\xi') > 0$. Therefore, optimizing the loss function will drive $\xi$ towards $\xi'$ until $\{\check{\xi}_{12}\} = \emptyset$, such that $F(\xi) \supseteq \{a_{12}\}$. Similarly, suppose that $\{\check{\xi}_{13}\} = (\{a_{00}, a_{11}, a_{22}, a_{33}, a_{12}, a_{13}, a_{23}\}/(\{a_{00}, a_{11}, a_{22}, a_{33}, a_{12}, a_{13}, a_{23}\} \cap I(\xi))) \cap \{a_{13}\}$ represents the task-relevant information in $a_{13}$ that is not included in $\xi$; and $\{\check{\xi}_{23}\} = (\{a_{00}, a_{11}, a_{22}, a_{33}, a_{12}, a_{13}, a_{23}\}/(\{a_{00}, a_{11}, a_{22}, a_{33}, a_{12}, a_{13}, a_{23}\} \cap I(\xi))) \cap \{a_{23}\}$ represents the task-relevant information in $a_{23}$ that is not included in $\xi$. Following the above procedure, we conclude that if $\{\check{\xi}_{13}\} \neq \emptyset$ and $\beta < \frac{I(y; \check{\xi}_{13})}{F(\xi,y)(I(v_1; \check{\xi}_{13}) + r_2 I(v_2; \check{\xi}_{13}))}$, the optimization will update $\xi$ until $\{\check{\xi}_{13}\} = \emptyset$, leading to $F(\xi) \supseteq \{a_{13}\}$; if $\{\check{\xi}_{23}\} \neq \emptyset$ and $\beta < \frac{I(y; \check{\xi}_{23})}{F(\xi,y)(r_1 I(v_1; \check{\xi}_{23}) + r_2 I(v_2; \check{\xi}_{23}))}$, the optimization will update $\xi$ until $\{\check{\xi}_{23}\} = \emptyset$, leading to $F(\xi) \supseteq \{a_{23}\}$.

Finally, we analyze under which condition $\xi$ can include all the task-relevant information shared by the three modalities. Let $\{\check{\xi}_0\} = (\{a_{00}, a_{11}, a_{22}, a_{33}, a_{12}, a_{13}, a_{23}\}/(\{a_{00}, a_{11}, a_{22}, a_{33}, a_{12}, a_{13}, a_{23}\} \cap I(\xi))) \cap \{a_{00}\}$ represent the task-relevant information in $a_{00}$ that is not included in $\xi$. It is obvious:

$$\{\check{\xi}_0\} \subset F(y), \{\check{\xi}_0\} \cap F(\xi) = \emptyset,$$
$$\{\check{\xi}_0\} \cap \{a_{ij}\} = \emptyset, \quad \forall a_{ij} \in \{a\}/\{a_{00}\},$$
$$\{\check{\xi}_0\} \cap F(v_l) = \emptyset, \quad \forall l \in \{1, 2, 3\}$$

If $\{\check{\xi}_0\} \neq \emptyset$, let $\xi' = \{\xi, \check{\xi}_0\}$, and $\ell(\xi) - \ell(\xi')$ can be written as:

$$\ell(\xi) - \ell(\xi') = -\widehat{I}(\xi; y) + \beta(I(\xi; v_1) + r_1 I(\xi; v_2) + r_2 I(\xi; v_3))$$
$$- \Big(-\widehat{I}(\xi, \check{\xi}_0; y) + \beta(I(\xi, \check{\xi}_0; v_1) + r_1 I(\xi, \check{\xi}_0; v_2) + r_2 I(\xi, \check{\xi}_0; v_3))\Big)$$
$$= \widehat{I}(\xi, \check{\xi}_0; y) - \widehat{I}(\xi; y) + \beta\Big(I(\xi; v_1) - I(\xi, \check{\xi}_0; v_1)$$
$$+ r_1(I(\xi; v_2) - I(\xi, \check{\xi}_0; v_2)) + r_2(I(\xi; v_3) - I(\xi, \check{\xi}_0; v_3))\Big)$$

Using properties in Appendix A, we have:

$$\widehat{I}(\xi, \check{\xi}_0; y) - \widehat{I}(\xi; y) = \frac{I(\xi, \check{\xi}_0; y)}{F(\xi) \cup \underbrace{(F(y) \cup F(\check{\xi}_0))}_{=F(y) \text{ as } \{\check{\xi}_0\} \subset F(y)}} - \frac{I(\xi; y)}{F(\xi) \cup F(y)}$$
$$= \frac{I(\xi, \check{\xi}_0; y)}{F(\xi, y)} - \frac{I(\xi; y)}{F(\xi, y)}$$
$$= \frac{\overbrace{I(y; \check{\xi}_0|\xi)}^{\{\check{\xi}_0\} \cap \{\xi\} = \emptyset}}{F(\xi, y)}$$
$$= \frac{I(y; \check{\xi}_0)}{F(\xi, y)} > 0$$

$$I(\xi; v_1) - I(\xi, \check{\xi}_0; v_1) = - \overbrace{I(v_1; \check{\xi}_0|\xi)}^{\{\check{\xi}_0\} \cap \{\xi\} = \emptyset} \tag{76}$$
$$= -I(v_1; \check{\xi}_0) < 0$$

Similarly, we obtain that $I(\xi; v_2) - I(\xi, \check{\xi}_0; v_2) = -I(v_2; \check{\xi}_0)$ and $I(\xi; v_3) - I(\xi, \check{\xi}_0; v_3) = -I(v_3; \check{\xi}_0)$.
Thus:

$$\begin{aligned}
\ell(\xi) - \ell(\xi') &= \hat{I}(\xi, \check{\xi}_0; y) - \hat{I}(\xi; y) + \beta \Big( I(\xi; v_1) - I(\xi, \check{\xi}_0; v_1) \\
&\quad + r_1(I(\xi; v_2) - I(\xi, \check{\xi}_0; v_2)) + r_2(I(\xi; v_3) - I(\xi, \check{\xi}_0; v_3)) \Big) \\
&= \frac{I(y; \check{\xi}_0)}{F(\xi, y)} - \beta(I(v_1; \check{\xi}_0) + r_1 I(v_2; \check{\xi}_0) + r_2 I(v_3; \check{\xi}_0))
\end{aligned} \tag{77}$$

When $\beta < \frac{I(y; \check{\xi}_0)}{F(\xi, y)(I(v_1; \check{\xi}_0) + r_1 I(v_2; \check{\xi}_0) + r_2 I(v_3; \check{\xi}_0))}$, $\ell(\xi) - \ell(\xi') > 0$. Therefore, optimizing the loss function will drive $\xi$ toward $\xi'$ until $\{\check{\xi}_0\} = \emptyset$, such that $F(\xi) \supseteq \{a_{00}\}$.

Put together, the optimization procedure ensures $F(\xi) \supseteq \{a_{00}, a_{11}, a_{22}, a_{33}, a_{12}, a_{13}, a_{23}\}$ when:

$$\beta < UB_\beta := min(UB_\beta^1, UB_\beta^2, UB_\beta^3, UB_\beta^4, UB_\beta^5, UB_\beta^6, UB_\beta^7). \tag{78}$$

where $UB_\beta^1 = \frac{I(y; \check{\xi}_i)}{F(\xi, y) I(v_1; \check{\xi}_i)}$, $UB_\beta^2 = \frac{I(y; \check{\xi}_2)}{r_1 F(\xi, y) I(v_2; \check{\xi}_2)}$, $UB_\beta^3 = \frac{I(y; \check{\xi}_3)}{r_2 F(\xi, y) I(v_3; \check{\xi}_3)}$, $UB_\beta^4 = \frac{I(y; \check{\xi}_{12})}{F(\xi, y)(I(v_1; \check{\xi}_{12}) + r_1 I(v_2; \check{\xi}_{12}))}$, $UB_\beta^5 = \frac{I(y; \check{\xi}_{13})}{F(\xi, y)(I(v_1; \check{\xi}_{13}) + r_2 I(v_2; \check{\xi}_{13}))}$, $UB_\beta^6 = \frac{I(y; \check{\xi}_{23})}{F(\xi, y)(r_1 I(v_1; \check{\xi}_{23}) + r_2 I(v_2; \check{\xi}_{23}))}$, and $UB_\beta^7 = \frac{I(y; \check{\xi}_0)}{F(\xi, y)(I(v_1; \check{\xi}_0) + r_1 I(v_2; \check{\xi}_{12}) + r_2 I(v_3; \check{\xi}_0))}$.

We complete the proof by proving that $M_u^2 = \frac{1}{(1 + r_1 + r_2)(\sum_{i=1}^3 H(v_i) - \frac{2}{3} \sum_{1 \le i < j \le 3} I(v_i; v_j))}$ is a lower bound of $UB_\beta$ in Lemma D.4 below. That is, when $\beta < M_u^2$, the optimization procedure guarantees $F(\xi) \supseteq \{a_{00}, a_{11}, a_{22}, a_{33}, a_{12}, a_{13}, a_{23}\}$. $\qquad \square$

**Lemma D.4.** $UB_\beta$ in Equation (78) satisfies: $UB_\beta > M_u^2$, where $M_u^2 = \frac{1}{(1 + r_1 + r_2)(\sum_{i=1}^3 H(v_i) - \frac{2}{3} \sum_{1 \le i < j \le 3} I(v_i; v_j))}$.
$H(\cdot)$ and $I(\cdot; \cdot)$ can be estimated using MINE (Belghazi et al., 2018) (see Appendix E).

*Proof.* As shown in Equation (78), $UB_\beta^1 = \frac{I(y; \check{\xi}_1)}{F(\xi, y) I(v_1; \check{\xi}_1)}$. By property **v** in Properties A.1, $UB_\beta^1$ can be simplified as:

$$\begin{aligned}
UB_\beta^1 &= \frac{I(y; \check{\xi}_1)}{I(v_1; \check{\xi}_1) F(\xi, y)} \\
&= \frac{\overbrace{H(\check{\xi}_1)}^{=I(y; \check{\xi}_1), \therefore \{\check{\xi}_1\} \subseteq \{y\}}}{\underbrace{H(\check{\xi}_1)}_{=I(v_1; \check{\xi}_1), \therefore \{\check{\xi}_1\} \subseteq \{v_1\}} F(\xi, y)} \\
&= \frac{1}{F(\xi, y)}
\end{aligned}$$

Similarly, we have $UB_\beta^2 = \frac{I(y; \check{\xi}_2)}{r_1 F(\xi, y) I(v_2; \check{\xi}_2)} = \frac{1}{r_1 F(\xi, y)}$ and $UB_\beta^3 = \frac{I(y; \check{\xi}_3)}{r_3 F(\xi, y) I(v_3; \check{\xi}_3)} = \frac{1}{r_2 F(\xi, y)}$.

$UB_\beta^4$ can be simplified as:

$$UB_\beta^4 = \frac{I(y;\check{\xi}_{12})}{F(\xi,y)(I(v_1;\check{\xi}_{12}) + r_1 I(v_2;\check{\xi}_{12}))}$$

$$= \frac{\overbrace{H(\check{\xi}_{12})}^{=I(y;\check{\xi}_{12}),\because\{\check{\xi}_{12}\}\subseteq\{y\}}}{F(\xi,y)(\underbrace{H(\check{\xi}_{12})}_{=I(v_1;\check{\xi}_{12}),\because\{\check{\xi}_{12}\}\subseteq\{v_1\}} + r_1 \underbrace{H(\check{\xi}_{12})}_{=I(v_2;\check{\xi}_{12}),\because\{\check{\xi}_{12}\}\subseteq\{v_2\}})} \tag{79}$$

$$= \frac{1}{(1+r_1)F(\xi,y)}$$

Similarly, we have $UB_\beta^5 = \frac{I(y;\check{\xi}_{13})}{F(\xi,y)(I(v_1;\check{\xi}_{13})+r_2 I(v_2;\check{\xi}_{13}))} = \frac{1}{(1+r_2)F(\xi,y)}$, and $UB_\beta^6 = \frac{I(y;\check{\xi}_{23})}{F(\xi,y)(r_1 I(v_1;\check{\xi}_{23})+r_2 I(v_2;\check{\xi}_{23}))} = \frac{1}{(r_1+r_2)F(\xi,y)}$.

$UB_\beta^7$ can be simplified as:

$$UB_\beta^7 = \frac{I(y;\check{\xi}_0)}{F(\xi,y)(I(v_1;\check{\xi}_0) + r_1 I(v_2;\check{\xi}_0) + r_2 I(v_3;\check{\xi}_0))}$$

$$= \frac{\overbrace{H(\check{\xi}_0)}^{=I(y;\check{\xi}_0),\because\{\check{\xi}_0\}\subseteq\{y\}}}{F(\xi,y)(\underbrace{H(\check{\xi}_0)}_{=I(v_1;\check{\xi}_0),\because\{\check{\xi}_0\}\subseteq\{v_1\}} + r_1 \underbrace{H(\check{\xi}_0)}_{=I(v_2;\check{\xi}_0),\because\{\check{\xi}_0\}\subseteq\{v_2\}} + r_2 \underbrace{H(\check{\xi}_0)}_{=I(v_3;\check{\xi}_0),\because\{\check{\xi}_0\}\subseteq\{v_3\}})}$$

$$= \frac{1}{(1+r_1+r_2)F(\xi,y)}$$

Therefore, for $\forall r_1, r_2 > 0$, we have:

$$\frac{1}{(1+r_1+r_2)F(\xi,y)} < \min(\frac{1}{F(\xi,y)}, \frac{1}{r_1 F(\xi,y)}, \frac{1}{r_2 F(\xi,y)}, \frac{1}{(1+r_1)F(\xi,y)}, \frac{1}{(1+r_2)F(\xi,y)}, \frac{1}{(r_1+r_2)F(\xi,y)}),$$

$$= \min(UB_\beta^1, UB_\beta^2, UB_\beta^3, UB_\beta^4, UB_\beta^5, UB_\beta^6, UB_\beta^7)$$

$$\implies UB_\beta = \frac{1}{(1+r_1+r_2)F(\xi,y)}$$

$$> \frac{1}{(1+r_1+r_2)F(v_1,v_2,v_3)}$$

$$= \frac{1}{(1+r_1+r_2)(H(v_1)+H(v_2)+H(v_3)-I(v_1;v_2)-I(v_1;v_3)-I(v_2;v_3)+I(v_1;v_2;v_3))}$$

For the term $I(v_1;v_2) + I(v_1;v_3) + I(v_2;v_3) - I(v_1;v_2;v_3)$, we have:

$$I(v_1;v_2) + I(v_1;v_3) + I(v_2;v_3) - I(v_1;v_2;v_3) < I(v_1;v_2) + I(v_1;v_3) + I(v_2;v_3) - I(v_1;v_2) = I(v_1;v_3) + I(v_2;v_3),$$

$$I(v_1;v_2) + I(v_1;v_3) + I(v_2;v_3) - I(v_1;v_2;v_3) < I(v_1;v_2) + I(v_1;v_3) + I(v_2;v_3) - I(v_1;v_3) = I(v_1;v_2) + I(v_2;v_3),$$

$$I(v_1;v_2) + I(v_1;v_3) + I(v_2;v_3) - I(v_1;v_2;v_3) < I(v_1;v_2) + I(v_1;v_3) + I(v_2;v_3) - I(v_2;v_3) = I(v_1;v_2) + I(v_1;v_3).$$

To calculate $I(v_1;v_2) + I(v_1;v_3) + I(v_2;v_3) - I(v_1;v_2;v_3)$, we sum up the individual inequalities, yielding:

$$I(v_1;v_2) + I(v_1;v_3) + I(v_2;v_3) - I(v_1;v_2;v_3) < \frac{1}{3}(I(v_1;v_3) + I(v_2;v_3) + I(v_1;v_2) + I(v_2;v_3) + I(v_1;v_2) + I(v_1;v_3))$$

$$= \frac{2}{3}\sum_{1\le i<j\le 3} I(v_i;v_j)$$

We then have:

$$
\begin{aligned}
UB_\beta &> \frac{1}{(1 + r_1 + r_2)(H(v_1) + H(v_2) + H(v_3) - I(v_1; v_2) - I(v_1; v_3) - I(v_2; v_3) + I(v_1; v_2; v_3))} \\
&> \frac{1}{(1 + r_1 + r_2)(\sum_{i=1}^{3} H(v_i) - \frac{2}{3} \sum_{1 \le i < j \le 3} I(v_i; v_j))} = M_u^2
\end{aligned}
$$

This completes the proof. $\qquad\square$

**Lemma D.5** (**Exclusiveness of superfluous information for three modalities**). *Under Assumption D.1, the objective function in Equation (69) is optimized when:*

$$
F(\xi) \subseteq \{a_{00}, a_{11}, a_{22}, a_{33}, a_{12}, a_{13}, a_{23}\} \tag{80}
$$

*Proof.* We begin by analyzing the change in the loss function of our optimization after adding modality-specific superfluous information to $\xi$. Let $\hat{\xi}_1$ represent $v_1$-specific superfluous information that is not incorporated into $\xi$. Obviously:

$$
\begin{aligned}
&\{\hat{\xi}_1\} \notin \{a_{00}, a_{11}, a_{22}, a_{33}, a_{12}, a_{13}, a_{23}\}, \{\hat{\xi}_1\} \subset F(v_1), I(\hat{\xi}_1; v_1) > 0, \\
&I(\hat{\xi}_1; y) = 0, \{\xi\} \cap \{\hat{\xi}_1\} = \emptyset, \{v_2\} \cap \{\hat{\xi}_1\} = \emptyset, \{v_3\} \cap \{\hat{\xi}_1\} = \emptyset.
\end{aligned} \tag{81}
$$

Let $\ddot{\xi} = \{\xi, \hat{\xi}_1\}$. The difference in loss function between $\xi$ and $\ddot{\xi}$ is computed as:

$$
\begin{aligned}
\ell(\ddot{\xi}) - \ell(\xi) &= -\widehat{I}(\ddot{\xi}; y) + \beta(I(\ddot{\xi}; v_1) + r_1 I(\ddot{\xi}; v_2) + r_2 I(\ddot{\xi}; v_3)) \\
&\quad - \left( -\widehat{I}(\xi; y) + \beta(I(\xi; v_1) + r_1 I(\xi; v_2) + r_2 I(\xi; v_3)) \right) \\
&= (\widehat{I}(\xi; y) - \widehat{I}(\xi, \hat{\xi}_1; y)) + \beta\Big( \widehat{I}(\hat{\xi}_1, \xi; v_1) - \widehat{I}(\xi; v_1) \\
&\quad + r_1(\widehat{I}(\hat{\xi}_1, \xi; v_2) - \widehat{I}(\xi; v_2)) + r_2(\widehat{I}(\hat{\xi}_1, \xi; v_3) - \widehat{I}(\xi; v_3)) \Big),
\end{aligned} \tag{82}
$$

Here we have:

$$
\begin{aligned}
\widehat{I}(\xi; y) - \widehat{I}(\hat{\xi}_1, \xi; y) &= \frac{I(\xi; y)}{F(\xi, y)} - \frac{I(\hat{\xi}_1, \xi; y)}{F(\hat{\xi}_1, \xi, y)} \\
&= \frac{I(\xi; y) + \overbrace{I(y; \hat{\xi}_1 | \xi)}^{=0 \; \because I(\hat{\xi}_1, y)=0}}{F(\xi, y)} - \frac{I(\hat{\xi}_1, \xi; y)}{F(\hat{\xi}_1, \xi, y)} \\
&= \frac{I(\hat{\xi}_1, \xi; y)}{F(\xi, y)} - \frac{I(\hat{\xi}_1, \xi; y)}{F(\hat{\xi}_1, \xi, y)} \\
&= \frac{I(\hat{\xi}_1, \xi; y)}{F(\xi, y)} - \frac{I(\hat{\xi}_1, \xi; y)}{\underbrace{F(\hat{\xi}_1) + F(\xi, y)}_{\because \hat{\xi}_1 \perp \{\xi, y\}}} \\
&> 0
\end{aligned} \tag{83}
$$

$$
I(\hat{\xi}_1, \xi; v_1) - I(\xi; v_1) = \overbrace{I(v_1; \hat{\xi}_1 | \xi)}^{\{\hat{\xi}_1\} \cap F(\xi) = \emptyset} \tag{84}
$$
$$
= I(v_1; \hat{\xi}_1) > 0
$$

$$I(\hat{\xi}_1, \xi; v_2) - I(\xi; v_2) = \overbrace{I(v_2; \hat{\xi}_1 | \xi)}^{\{\hat{\xi}_1\} \cap F(v_2) = \emptyset} \tag{85}$$
$$= 0$$

$$I(\hat{\xi}_1, \xi; v_3) - I(\xi; v_3) = 0 \tag{86}$$

Thus, we have $\ell(\ddot{\xi}) - \ell(\xi) > 0$, if $\{\hat{\xi}_1\} \neq \emptyset$. For superfluous information $\hat{\xi}_2$ specific to $v_2$ and $\hat{\xi}_3$ specific to $v_3$, we arrive at the same conclusion. Next, we analyze the change in the loss function of our optimization after adding superfluous information shared by two modalities to $\xi$. Specifically, let $\hat{\xi}_{12}$ represent the superfluous information that is shared between modalities $v_1$ and $v_2$, and not incorporated into $\xi$. We have:

$$\hat{\xi}_{12} \notin \{a_{00}, a_{11}, a_{22}, a_{33}, a_{12}, a_{13}, a_{23}\}, \{\hat{\xi}_{12}\} \subset F(v_1), \{\hat{\xi}_{12}\} \subset F(v_2),$$
$$I(\hat{\xi}_{12}; v_1) > 0, I(\hat{\xi}_{12}; v_2) > 0, \tag{87}$$
$$I(\hat{\xi}_{12}; y) = 0, \{\xi\} \cap \{\hat{\xi}_{12}\} = \emptyset, \{v_3\} \cap \{\hat{\xi}_{12}\} = \emptyset.$$

Let $\ddot{\xi} = \{\xi, \hat{\xi}_{12}\}$. The difference in loss function between $\xi$ and $\ddot{\xi}$ is computed as:

$$\begin{aligned} \ell(\ddot{\xi}) - \ell(\xi) &= -\widehat{I}(\ddot{\xi}; y) + \beta(I(\ddot{\xi}; v_1) + r_1 I(\ddot{\xi}; v_2) + r_2 I(\ddot{\xi}; v_3)) \\ &\quad - \left( -\widehat{I}(\xi; y) + \beta(I(\xi; v_1) + r_1 I(\xi; v_2) + r_2 I(\xi; v_3)) \right) \\ &= (\widehat{I}(\xi; y) - \widehat{I}(\xi, \hat{\xi}_{12}; y)) + \beta\left( \widehat{I}(\hat{\xi}_{12}, \xi; v_1) - \widehat{I}(\xi; v_1) \right. \\ &\quad \left. + r_1(\widehat{I}(\hat{\xi}_{12}, \xi; v_2) - \widehat{I}(\xi; v_2)) + r_2(\widehat{I}(\hat{\xi}_{12}, \xi; v_3) - \widehat{I}(\xi; v_3)) \right) \end{aligned} \tag{88}$$

Here we have:

$$\begin{aligned} \widehat{I}(\xi; y) - \widehat{I}(\hat{\xi}_{12}, \xi; y) &= \frac{I(\xi; y)}{F(\xi, y)} - \frac{I(\hat{\xi}_{12}, \xi; y)}{F(\hat{\xi}_{12}, \xi, y)} \\ &= \frac{I(\xi; y) + \overbrace{I(y; \hat{\xi}_{12} | \xi)}^{=0 \, \because I(\hat{\xi}_{12}, y) = 0}}{F(\xi, y)} - \frac{I(\hat{\xi}_{12}, \xi; y)}{F(\hat{\xi}_{12}, \xi, y)} \\ &= \frac{I(\hat{\xi}_{12}, \xi; y)}{F(\xi, y)} - \frac{I(\hat{\xi}_{12}, \xi; y)}{F(\hat{\xi}_{12}, \xi, y)} \\ &= \frac{I(\hat{\xi}_{12}, \xi; y)}{F(\xi, y)} - \frac{I(\hat{\xi}_{12}, \xi; y)}{\underbrace{F(\hat{\xi}_{12}) + F(\xi, y)}_{\because \hat{\xi}_{12} \perp \{\xi, y\}}} \\ &> 0 \end{aligned} \tag{89}$$

$$I(\hat{\xi}_{12}, \xi; v_1) - I(\xi; v_1) = \overbrace{I(v_1; \hat{\xi}_{12} | \xi)}^{\{\hat{\xi}_{12}\} \cap F(\xi) = \emptyset} \tag{90}$$
$$= I(v_1; \hat{\xi}_{12}) > 0$$

$$I(\hat{\xi}_{12}, \xi; v_2) - I(\xi; v_2) = I(v_2; \hat{\xi}_{12}) > 0 \tag{91}$$

$$I(\hat{\xi}_{12}, \xi; v_3) - I(\xi; v_3) = \overbrace{I(v_3; \hat{\xi}_{12}|\xi)}^{\{\hat{\xi}_{12}\} \cap F(v_3) = \emptyset} \tag{92}$$
$$= 0$$

Thus, we have $\ell(\ddot{\xi}) - \ell(\xi) > 0$, if $\{\hat{\xi}_{12}\} \neq \emptyset$. For superfluous information $\hat{\xi}_{13}$ shared between modalities $v_1$ and $v_3$, as well as $\hat{\xi}_{23}$ shared between modalities $v_2$ and $v_3$, we arrive at the same conclusion. Finally, we analyze the change in the loss function of our optimization after adding superfluous information shared by all three modalities to $\xi$. Let $\hat{\xi}_0$ represent the superfluous information shared by all three modalities and not incorporated into $\xi$. Then, we have:

$$\begin{aligned}
&\hat{\xi}_0 \notin \{a_{00}, a_{11}, a_{22}, a_{33}, a_{12}, a_{13}, a_{23}\}, \\
&\{\hat{\xi}_0\} \subset F(v_1), \{\hat{\xi}_0\} \subset F(v_2), \{\hat{\xi}_0\} \subset F(v_3), \\
&I(\hat{\xi}_0; v_1) > 0, I(\hat{\xi}_0; v_2) > 0, I(\hat{\xi}_0; v_3) > 0, \\
&I(\hat{\xi}_0; y) = 0, \{\xi\} \cap \{\hat{\xi}_0\} = \emptyset.
\end{aligned} \tag{93}$$

Let $\ddot{\xi} = \{\xi, \hat{\xi}_0\}$. The difference in loss function between $\xi$ and $\ddot{\xi}$ is computed as:

$$\begin{aligned}
\ell(\ddot{\xi}) - \ell(\xi) &= -\widehat{I}(\ddot{\xi}; y) + \beta(I(\ddot{\xi}; v_1) + r_1 I(\ddot{\xi}; v_2) + r_2 I(\ddot{\xi}; v_3)) \\
&\quad - \left( -\widehat{I}(\xi; y) + \beta(I(\xi; v_1) + r_1 I(\xi; v_2) + r_2 I(\xi; v_3)) \right) \\
&= (\widehat{I}(\xi; y) - \widehat{I}(\xi, \hat{\xi}_0; y)) + \beta\Big( \widehat{I}(\hat{\xi}_0, \xi; v_1) - \widehat{I}(\xi; v_1) \\
&\quad + r_1(\widehat{I}(\hat{\xi}_0, \xi; v_2) - \widehat{I}(\xi; v_2)) + r_2(\widehat{I}(\hat{\xi}_0, \xi; v_3) - \widehat{I}(\xi; v_3)) \Big)
\end{aligned} \tag{94}$$

Here we have:

$$\begin{aligned}
\widehat{I}(\xi; y) - \widehat{I}(\hat{\xi}_0, \xi; y) &= \frac{I(\xi; y)}{F(\xi, y)} - \frac{I(\hat{\xi}_0, \xi; y)}{F(\hat{\xi}_0, \xi, y)} \\
&= \frac{I(\xi; y) + \overbrace{I(y; \hat{\xi}_0|\xi)}^{=0 \,\because I(\hat{\xi}_0, y) = 0}}{F(\xi, y)} - \frac{I(\hat{\xi}_0, \xi; y)}{F(\hat{\xi}_0, \xi, y)} \\
&= \frac{I(\hat{\xi}_0, \xi; y)}{F(\xi, y)} - \frac{I(\hat{\xi}_0, \xi; y)}{F(\hat{\xi}_0, \xi, y)} \\
&= \frac{I(\hat{\xi}_0, \xi; y)}{F(\xi, y)} - \frac{I(\hat{\xi}_0, \xi; y)}{\underbrace{F(\hat{\xi}_0) + F(\xi, y)}_{\because \hat{\xi}_0 \perp \{\xi, y\}}} \\
&> 0
\end{aligned} \tag{95}$$

$$I(\hat{\xi}_0, \xi; v_1) - I(\xi; v_1) = \overbrace{I(v_1; \hat{\xi}_0|\xi)}^{\{\hat{\xi}_0\} \cap F(\xi) = \emptyset} \tag{96}$$
$$= I(v_1; \hat{\xi}_0) > 0$$

$$I(\hat{\xi}_0, \xi; v_2) - I(\xi; v_2) = I(v_2; \hat{\xi}_0) > 0 \tag{97}$$

$$I(\hat{\xi}_0, \xi; v_3) - I(\xi; v_3) = I(v_3; \hat{\xi}_0) > 0 \tag{98}$$

Thus $\ell(\ddot{\xi}) - \ell(\xi) > 0$, if $\{\hat{\xi}_0\} \neq \emptyset$. Put together, the optimization procedure continues until $\xi$ does not encompass superfluous information, specific to or shared by $v_1$, $v_2$, and $v_3$. That is, $F(\xi) \subseteq \{a_{00}, a_{11}, a_{22}, a_{33}, a_{12}, a_{13}, a_{23}\}$. This completes the proof. $\square$

**Proposition D.6** (**Achievability of optimal MIB for three modalities**). *Lemma D.3, and Lemma D.5 jointly demonstrate that the optimal MIB $\xi_{opt-three}$ is achievable through optimization of Equation* (69) *with $\beta \in (0, M_u^2]$.*

*Proof.* From Lemma D.3 and Lemma D.5, we have $F(\xi) \supseteq \{a_{00}, a_{11}, a_{22}, a_{33}, a_{12}, a_{13}, a_{23}\}$ if $\beta \in (0, M_u^2]$, and $F(\xi) \subseteq \{a_{00}, a_{11}, a_{22}, a_{33}, a_{12}, a_{13}, a_{23}\}$, respectively. Thus, $F(\xi) = \{a_{00}, a_{11}, a_{22}, a_{33}, a_{12}, a_{13}, a_{23}\}$, which corresponds to $\xi_{opt-three}$ in Definition D.2. $\square$

To expedite the training process, we can also set $M_u^2 = \frac{1}{5(\sum_{i=1}^3 H(v_i) - \frac{2}{3}\sum_{1 \leq i < j \leq 3} I(v_i; v_j))}$ as an upper bound and $M_l^2 := \frac{1}{5(\sum_{i=1}^3 H(v_i))}$ as a lower bound for $\beta$, resulting in $\beta \in [M_l^2, M_u^2]$.

## E. Estimation of Mutual Information and Information Entropy

We apply the Mutual Information Neural Estimation (MINE) method (Belghazi et al., 2018) to estimate the information entropy of each data modality and the mutual information between data modalities. Given two modalities $X$ and $Z$, MINE employs a neural network, implemented as a two-layer Multi-Layer Perceptron (MLP) network with ReLU activation function (Belghazi et al., 2018), to learn a set of functions $\{T_\theta\}_{\theta \in \Theta}$. Each function $T_\theta : X \times Z \to \mathbb{R}$ maps sample pairs to real values, enabling mutual information estimation as:

$$I(X; Z) = \sup_{\theta \in \Theta} \mathbb{E}_{P_{XZ}}[T_\theta] - \log \mathbb{E}_{P_X \otimes P_Z}[e^{T_\theta}]. \tag{99}$$

Here, $\mathbb{E}_{P_{XZ}}[T_\theta]$ represents the expected value of $T_\theta$ calculated using sample pairs from the joint distribution $P_{XZ}$, and $\mathbb{E}_{P_X \otimes P_Z}[e^{T_\theta}]$ represents the expected value of $T_\theta$ calculated using sample pairs from the product of marginal distribution $P_X \otimes P_Z$. The joint distribution $P_{XZ}$ is approximated using matched sample pairs $(X, Z)$, while $P_X \otimes P_Z$ is approximated using perturbed pairs $(X, Z')$, where $Z'$ is obtained by shuffling $Z$. The information entropy $H(X)$ is computed as the mutual information of $X$ with itself:

$$H(X) = I(X; X) \tag{100}$$

Specifically, in this case, $Z$ and $Z'$ are replaced by $X$ and $X'$, where $X'$ is obtained by shuffling $X$.

## F. Synthetic Data

Following (Xue et al., 2023), we simulate pairs of Gaussian observations and task labels, $x_1 \in \mathbb{R}^{d_1}$, $x_2 \in \mathbb{R}^{d_2}$; $y$, where $x_1$ and $x_2$ represent observations from two modalities with dimensionalities $d_1$ and $d_2$, respectively, and $y \in \{0, 1\}$ represents the corresponding binary label. The feature vectors of $x_1$ and $x_2$ are defined as:

$$x_1 = [b_0; b_1; a_0; a_1], x_2 = [b_0; b_2; a_0; a_2], \tag{101}$$

where

- $a_0 \in \mathbb{R}^{d_0} \sim \mathcal{N}(0, I_{d_0})$ denotes consistent, task-relevant information shared between the modalities;

- $a_1 \in \mathbb{R}^{d_{11}} \sim \mathcal{N}(0, I_{d_{11}})$ and $a_2 \in \mathbb{R}^{d_{21}} \sim \mathcal{N}(0, I_{d_{21}})$ represent modality-specific, task-relevant information;

- $b_0 \in \mathbb{R}^{d_0'} \sim \mathcal{N}(0, I_{d_0'})$ is consistent, superfluous information;

- $b_1 \in \mathbb{R}^{d_{12}} \sim \mathcal{N}(0, I_{d_{12}})$ and $b_2 \in \mathbb{R}^{d_{22}} \sim \mathcal{N}(0, I_{d_{22}})$ are modality-specific, superfluous information.

Here, $\mathcal{N}(0, I_d)$ denotes a multivariate Gaussian distribution with mean 0 and identity covariance matrix $I$ of dimensionality $d$. The dimensions satisfy:

$$d_1 = d_0 + d_0' + d_{11} + d_{12}, d_2 = d_0 + d_0' + d_{21} + d_{22} \tag{102}$$

The label $y \in \{0, 1\}$ is generated using a Dirac function $\Delta$, which depends solely on the task-relevant information $a_0, a_1$, and $a_2$:

$$y := \Delta(\langle \delta, [a_0; a_1; a_2] \rangle > 0) \tag{103}$$

where $\delta \in \mathbb{R}^{d_0+d_{11}+d_{21}} \sim \mathcal{N}(0, I_{d_0+d_{11}+d_{21}})$ is a randomly sampled vector serving as a separating hyperplane, and $\langle \cdot, \cdot \rangle$ denotes the inner product operation.

By adjusting $d_0$, $d_{11}$, and $d_{21}$, we can control the distribution of task-relevant information across the two modalities, enabling the simulation of imbalanced task-relevant information. Specifically, as illustrated in Figure 3, we simulate three SIM datasets (SIM-{I-III}) to be used in three experimental cases, respectively (see Section 6.2). Firstly, for all cases, we set $d_0 = d_0' = 200$. For SIM-I used in **case i**, we set $d_{11}(500) \gg d_{21}(100)$ so that $a_1$ has a significantly greater impact on determining $y$, compared to $a_2$. This configuration implies that Modality I dominates Modality II in terms of task-relevant information. For SIM-II used in **case ii**, we switch the setting of $d_{11}$ and $d_{12}$, making Modality II dominant over Modality I. Finally, for SIM-III used in **case iii**, we set $d_{11} = d_{12} = 300$ to ensure both modalities contribute equally to task-relevant information.

## G. Detailed Dataset Description

**SIM.**  See Appendix F.

**CREMA-D.**  CREMA-D is an audio-visual dataset designed to study multimodal emotional expression and perception (Cao et al., 2014). It captures actors portraying six basic emotional states—happy, sad, anger, fear, disgust, and neutral—through facial expressions and speech.

**CMU-MOSI.**  CMU-MOSI (Zadeh et al., 2016) consists of 93 videos, from which 2,199 utterance are generated, each containing an image, audio, and language component. Each utterance is labeled with sentiment intensity ranging from -3 to 3.

**10x-hNB-{A-H}& 10x-hBC-{A-D}.**  The 10x-hNB-{A-H} datasets comprises eight datasets derived from healthy human breast tissues, while the 10x-hBC-{A-D} datasets contain four datasets from human breast cancer tissues (Xu et al., 2024b). As shown in Figure 6, each dataset corresponds to a tissue section and include gene expression and histology modalities. For each tissue section, gene expression profiles (i.e., gene read counts) are measured at fixed spatial spots across the section. During data preprocessing, genes detected in fewer than 10 spots are excluded, and raw gene expression counts are normalized by library size, log-transformed, and reduced to the 3,000 highly variable genes (HVGs) using the SCANPY package (Wolf et al., 2018; Li et al., 2024; Xu et al., 2024a; Du et al., 2025). The corresponding histology image is segmented into $32 \times 32$ region patches centered around each spatial spot, from which pathological patches are identified for anomaly detection. OMIB and baseline models are trained on the 10x-hBC-{A-H} datasets to learn multimodal representations of normal tissue regions within a compact hypersphere in the latent space. The trained models are then applied to the 10x-hBC-{A-D} datasets during inference.

## H. Detailed Network Architecture Implementation

**Modality-specific encoder.**  We implement the encoder as follows:

- The SIM datasets: A two-layer MLP with the GELU activation function, outputting 256-dimensional embeddings.

- The CREMA-D dataset: Both video and audio encoders use ResNet-18, producing 512-dimensional outputs.

- The CMU-MOSI dataset: Conv1D is employed for both the audio and visual modalities, while BERT is utilized for the textual modality, with all three encoders producing 512-dimensional embeddings.

- The 10x-hNB-{A-H}& 10x-hBC-{A-D} datasets: ResNet-18 and a two-layer graph convolutional network are used for the histology and gene expression modalities, respectively, with both encoders producing 256-dimensional embeddings.

Table 7: Overview of the experimental datasets.

| Dataset | Type | Number of samples (Anomaly proportion) |
|---------|------|----------------------------------------|
| SIM-{I-III} | Training | 9000 |
| SIM-{I-III} | Test | 1000 |
| CREMA-D | Training | 6,698 |
| CREMA-D | Test | 744 |
| MOSI | Training | 1281 |
| MOSI | Test | 685 |
| 10x-hNB-A | Training | 2364 |
| 10x-hNB-B | Training | 2504 |
| 10x-hNB-C | Training | 2224 |
| 10x-hNB-D | Training | 3037 |
| 10x-hNB-E | Training | 2086 |
| 10x-hNB-F | Training | 2801 |
| 10x-hNB-G | Training | 2694 |
| 10x-hNB-H | Training | 2473 |
| 10x-hBC-A | Test | 346 (12.43%) |
| 10x-hBC-B | Test | 295 (78.64%) |
| 10x-hBC-C | Test | 176 (27.84%) |
| 10x-hBC-D | Test | 306 (54.58%) |

**Task-relevant prediction head.** We implement task-relevant prediction head as follows:

- The SIM and CREMA-D datasets: The prediction head is implemented as a single linear layer (input X 512 X 100) followed by a softmax layer for classification, producing a $k$-dimensional output, where $k$ is the number of classification types. The TRB loss $L_{TRB}$ is cross-entropy;

- The CMU-MOSI dataset: The prediction head is implemented as a single linear layer MLP (input X 50 X 1), outputting a single real value. $L_{TRB}$ is mean squared error;

- The 10x-hNB-{A-H} and 10x-hBC-{A-D} datasets: The prediction head is implemented under the SVDD framework (Ruff et al., 2018; Xu et al., 2025) as a two-layer MLP (input X 256 X 256) with LeakyReLU activation functions, producing 256-dimensional latent multimodal representations. $L_{TRB}$ is defined as:

$$L_{TRB} = \frac{1}{N} \sum_{i=1}^{N} \|\hat{y} - c\|^2 + \lambda \cdot \mathcal{R}(\Theta),$$

$$c = \frac{1}{N} \sum_{i=1}^{N} \hat{y}, \qquad (104)$$

where $\hat{y}$ denotes the output of the prediction head, $c$ the center of the hypersphere, $\mathcal{R}(\Theta)$ the function that regularizes model parameters $\Theta$ for reducing model complexity and preventing model collapse, $\lambda$ is the regularization weight.

**Variational Autoencoder.** The VAE is implemented as two-layer MLP with two heads, outputting the $\mu$ and $\Sigma$, respectively.

**Cross-Attention Network.** For datasets with two modalities, the cross-attention is implemented as:

$$\xi = \text{Attn}\left([\zeta_1 \| \zeta_2]; W_Q, W_K, W_V\right) \qquad (105)$$

where $\text{Attn}$ represents the standard attention block, $W_Q$, $W_K$, and $W_V$ denote learnable projection matrices for queries, keys, and values respectively. The operator $\|$ represents concatenation along the feature dimension.

For datasets with three modalities, the cross-attention is extended as:

$$\xi = \text{Attn}\left([\zeta_1 \| \zeta_2 \| \zeta_3]; W_Q, W_K, W_V\right) \tag{106}$$

Finally, a Linear layer is applied to map $\xi$ back to the same dimensions as $\zeta_1$ and $\zeta_2$.

## I. Experimental Settings

All experiments are implemented using PyTorch (Paszke et al., 2019), with the following settings:

**SIM.**  We use the Adam optimizer with a learning rate of 1e-4 and train the model for 100 epochs. The dataset consists of 10,000 samples, split into training and test sets with a 9:1 ratio.

**CREMA-D.**  The model is trained using the SGD optimizer with a batch size of 64, momentum of 0.9, and weight decay of 1e-4. The learning rate is initialized at 1e-3 and decays by a factor of 0.1 every 70 epochs, reaching a final value of 1e-4. The dataset is divided into a training set containing 6,698 samples and a test set of 744 samples.

**CMU-MOSI.**  We employ the Adam optimizer with a learning rate of 1e-5. All other hyperparameters and settings follow (Mai et al., 2023). 2,199 utterances are extracted from the dataset, which are split into a training set (1,281 samples) and a test set (685 samples).

**10x-hNB-{A-H}& 10x-hBC-{A-D}.**  We use the Adam optimizer with a learning rate of 1e-4 and a weight decay of 0.1. The training batch size is set to 128. The final multimodal representation has a dimensionality of 256.

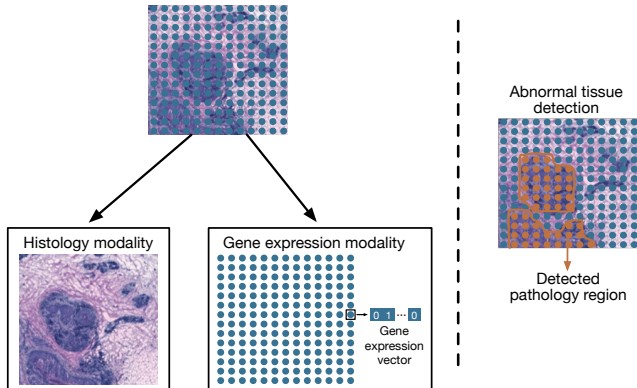

Figure 6: Genomic multi-modal applications. Genomic data can be divided into two modalities: the histology modality and the gene expression modality. The histology modality comprises tissue image, while the gene expression modality consists of gene expression vectors, where each spot corresponds to a vector composed of multiple gene expression values. These two modalities are spatially aligned through shared spatial information. By integrating and analyzing both modalities, abnormal regions within the tissue can be effectively detected.

## J. Benchmark Methods

Here, we briefly describe the eight benchmark methods used in this study. For non-MIB-based methods:

- Concat refers to simple concatenation of multi-modal features, which yet is the most widely used fusion approach.

- BiGated (Kiela et al., 2018) flexibly integrates information from different modalities through a gating mechanism.

- MISA (Hazarika et al., 2020) decomposes data into modality-invariant and modality-specific representations, using alignment and divergence constraints for better multimodal representation.

For MIB-based methods:

- deep IB (Wang et al., 2019) extends VIB to a multi-view setting, maximizing mutual information between labels and the joint representation while minimizing mutual information between each view's latent representation and the original data;

- MMIB-Cui (Cui et al., 2024) addresses the issues of modality noise and modality gap in multimodal named entity recognition (MNER) and multimodal relation extraction (MRE) by integrating the information bottleneck principle, thereby enhancing the semantic consistency between textual and visual information;

- MMIB-Zhang (Zhang et al., 2022) effectively controls the learning of multimodal representations by imposing mutual information constraints between different modality pairs, removing task-irrelevant information within single modalities while retaining relevant information, significantly improving performance in multimodal sentiment analysis;

- DMIB (Fang et al., 2024) effectively filters out irrelevant information and noise, while introducing a sufficiency loss to retain task-relevant information, demonstrating significant robustness in the presence of redundant data and noisy channels.

- E-MIB, L-MIB, and C-MIB (Mai et al., 2023) aim to learn effective multimodal and unimodal representations by maximizing task-relevant mutual information, eliminating modality redundancy, and filtering noise, while exploring the effects of applying MIB at different fusion stages.

## K. Evaluation Metrics

In Emotion Recognition, we use accuracy (Acc) as the evaluation metric. For Multimodal Sentiment Analysis, we use the mean absolute error (MAE) and Pearson's correlation coefficient (Corr) to evaluate the predicted scores against the true scores. Additionally, as sentiment intensity scores can be divided into positive and negative categories, F1-score and polarity accuracy (Acc-2) are also utilized to evaluate prediction results as a binary classification task. Additionally, the interval of $[-3, 3]$ contains seven integer scores to which each predicted score is neared to. This allows the using of categorical accuracy (Acc-7) to evaluate the prediction results. Finally, for the Anomalous Tissue Detection task, we evaluate performance using AUC score and F1-score. The AUC score is calculated by varying the anomaly threshold over all tissue regions' anomalous scores. To compute the F1-score, a threshold is first identified such that the number of regions exceeding it matches the true number of anomalous regions, after which the F1-score is computed for regions whose scores are above this threshold.

## L. Algorithmic workflow of *OMIB*

---

**Algorithm 1** Warm-up training

---

**Input:** Modality $v_k$, $k \in \{1, 2\}$, Maximum epochs $E_{max}$, Batch size $N$.

**Notation:** $Enc_k$: Unimodal encoder for modality $v_k$; $Dec_k$: Task-relevant prediction head for modality $v_k$; $z_k$: Latent representation of modality $v_k$; $e_k$: Stochastic Gaussian noises;

**Output:** $Enc_k$ and $Dec_k$.

1: Initialize $Enc_k$ and $Dec_k$, $\forall k \in \{1, 2\}$;
2: **while** $epoch < E_{max}$ **do**
3:      Sample a batch $\{v_k^i \mid i \in \{1, 2, \dots, N\}\}$ from each modality $k \in \{1, 2\}$;
4:      **for** each $i \in \{1, 2, \dots, N\}$ **do**
5:          **for** each modality $k \in \{1, 2\}$ **do**
6:              $z_k^i = Enc_k^i(v_k^i)$;
7:              $e_k^i \sim \mathcal{N}(0, I)$;
8:              $\hat{y}_k^i = Dec_k([z_k^i, e_k^i])$;
9:          **end for**
10:      **end for**
11:      Compute $L_{TRB_k}$ as in Equation (4) for each modality $k \in \{1, 2\}$;
12:      Update $Enc_k$ and $Dec_k$ using gradient descent;
13: **end while**
14: **return** $Enc_k$ and $Dec_k$

---

---

**Algorithm 2** Main training

---

**Input:** Modality $v_k$, Unimodal encoder $Enc_k$, Task-relevant prediction head $Dec_k$, $\forall k \in \{1, 2\}$, Maximum epochs $E_{max}$, Batch size $N$.

**Notation:** $VAE_k$: Variational encoder for modality $v_k$; $\zeta_k$: Latent representation of modality $v_k$ after reparameterization; $CAN$: Cross-attention network; $\widehat{Dec}$: OMF task-relevant prediction head; MINE: Mutual Information Neural Estimation (MINE); $\epsilon_k$: Standard Gaussian samples.

**Output:** $Enc_k$, $\forall k \in \{1, 2\}$, OMF ($VAE_k$, $\forall k \in \{1, 2\}$, $CAN$, and $\widehat{Dec}$).

1: **for** each modality $k \in \{1, 2\}$ **do**
2:     $H(v_k) = \text{MINE}(v_k, v_k)$;
3: **end for**
4: $I(v_1; v_2) = \text{MINE}(v_1, v_2)$;
5: Sample $\beta$ from the range $[M_l, M_u]$, where $M_l := \frac{1}{3(H(v_1)+H(v_2))}$, $M_u := \frac{1}{3(H(v_1)+H(v_2)-I(v_1;v_2))}$;
6: **while** $epoch < E_{max}$ **do**
7:     Sample a batch $\{v_k^i \mid i \in \{1, 2, \ldots, N\}\}$ from each modality $k \in \{1, 2\}$;
8:     **for** each $i \in \{1, 2, \ldots, N\}$ **do**
9:         **for** each modality $k \in \{1, 2\}$ **do**
10:             $z_k^i = Enc_k(v_k^i)$;
11:             $\mu_k^i, \Sigma_k^i = VAE_i(z_k^i)$;
12:             $\zeta_k^i = \mu_k^i + \Sigma_k^i \times \epsilon_i$;
13:         **end for**
14:         $\xi^i = CAN(\zeta_1^i, \zeta_2^i)$;
15:         **for** each modality $i \in \{1, 2\}$ **do**
16:             $\hat{y}_k^i = Dec_i([z_k^i, \xi^i])$;
17:         **end for**
18:         $\hat{y}^i = \widehat{Dec}(\xi^i)$;
19:         Adjust $r$ as defined in Equation (11);
20:     **end for**
21:     Compute $L_{OMF}$ as in Equation (10), and $L_{TRB_k}$ as in Equation (4) for each modality $i \in \{1, 2\}$;
22:     $L = L_{OMF} + L_{TRB_1} + L_{TRB_2}$;
23:     Update parameters of $Enc_k$, $VAE_k$, $CAN$, $Dec_k$, and $\widehat{Dec}$ using gradient descent;
24: **end while**
25: **return** $Enc_k$, $\forall k \in \{1, 2\}$, OMF ($VAE_k$, $\forall k \in \{1, 2\}$, $CAN$, and $\widehat{Dec}$)

---

