# OpenReview forum: "Learning Optimal Multimodal Information Bottleneck Representations"
_ICML.cc/2025/Conference — ICML 2025 poster_

### Official Review · Reviewer_BfWV · 2025-03-08

**Overall Recommendation:** 4

**Summary:**

The author introduces a theoretically guaranteed multimodal information bottleneck approach. This method dynamically adjusts the regularization weights of each modality by considering the varying degrees of task-relevant information across different modalities. Theoretically, the optimization objective proposed by the author is of a remarkably straightforward form, and the practical loss function serves as an upper bound to this theoretical objective, thereby ensuring the feasibility.


#### update after rebuttal:  I  don't  change   my assessment.

**Claims And Evidence:**

Most of the methods and propositions in the article are supported by theoretical foundations. However, Equation (3) raises some questions, which will be elucidated in the ‘Questions For Authors’

**Essential References Not Discussed:**

The article's citations are quite comprehensive.

**Experimental Designs Or Analyses:**

The simulated two-modality dataset is quite intriguing. However, it does not account for scenarios that require simultaneous decision-making across both modalities, such as sarcasm. For instance, consider $a, b \sim N(0, I_D)$, with the label $y = \Delta(a^T b > 0)$. In this case, $I(a; y) = I(b; y) = 0$, while $I(y; a, b) > 0$.

For the audio-visual dataset, it would be beneficial to include datasets other than CREMA-D, such as KS or VGGSound. Regarding CREMA-D, simply increasing the number of training epochs can enable a straightforward fusion method like concatenation to achieve a score close to 70 with Resnet18. Therefore, to rule out the possibility that the method merely converges quickly, it is advisable to either increase the number of learning epochs or introduce additional datasets.

**Methods And Evaluation Criteria:**

The information bottleneck framework proposed by the author addresses the overfitting predicament inherent in multimodal learning, whereas the adaptive regulation of regularization weights effectively addresses the challenges posed by imbalanced learning scenarios.

**Other Comments Or Suggestions:**

Since the derivations are all included in the appendix and only the conclusions are presented in the main text, Properties 1 can be removed from the main text.

**Other Strengths And Weaknesses:**

The article is logically structured, with clearly defined and reader-friendly symbols. The theoretical section is particularly detailed and rigorous.

**Questions For Authors:**

In the experiments, the non-MIB-based methods only include some basic approaches. I am curious about how they compare with newer methods:
Peng, Xiaokang, et al. "Balanced multimodal learning via on-the-fly gradient modulation." Proceedings of the IEEE/CVF conference on computer vision and pattern recognition. 2022.

Zhang, Xiaohui, et al. "Multimodal representation learning by alternating unimodal adaptation." Proceedings of the IEEE/CVF Conference on Computer Vision and Pattern Recognition. 2024.

I harbor a degree of skepticism towards the reasoning in Equation (3), which posits that concatenating $e_i$ enhances the model's learning by improving the signal-to-noise ratio. Typically, the introduction of signal-to-noise ratio considerations involves additive noise, such as ( z_i^{noise} = z_i + e_i ), rather than direct concatenation. Moreover, the ablation studies do not include relevant content to substantiate this.

**Relation To Broader Scientific Literature:**

The ingenious construction of the loss in the article and the proofs related to information theory in this section may provide valuable insights for future research in multimodal learning. A well-designed information bottleneck could also potentially benefit downstream tasks.

**Theoretical Claims:**

I carefully checked the proofs related to section 5.1 (Appendix B) and skimmed through the proofs associated with section 5.2 (Appendix C). The proof section is quite rigorous, and there is no obvious issues.

---

> ### Author Rebuttal · Authors · 2025-04-01
>
> **Experimental Designs Or Analyses:**
>
> **Q1.** Thank you for this insightful observation regarding synergistic interactions between modalities. In response, we conducted additional experiments using synthetic data with two modalities ($x_1,x_2$), where $x_1=[a_0;b_0]$, $x_2=[a_1;b_1]$, and $y=\Delta(a_0^Ta_1>0)$. All  $a_i$ and $b_i$ are sampled from the standard Gaussian, yielding 10,000 sample pairs. Theoretically, in this setting, the TRBs would be unable to distinguish between $a_i$ and $b_i$, leading to the encoder to randomly incorporate information. However, $L_{OMF}$ of the OMF block can capture the inter-modal synergic interactions and inform the encoders to extract task-relevant information. To validate this, we tested our method in two configurations: $L_{OMF}$ is either involved (*w-$L_{OMF}$*) or not (*wo-$L_{OMF}$*) in optimizing the modality encoders. For comparison, we also evaluated a single-modality baseline (*Single*) and the union of the two modalities (*Union*). The table below shows that *w-$L_{OMF}$* achieves the highest accuracy, followed by *Union* and *wo-$L_{OMF}$*, while *Single* performs near random guessing. These results align with our expectations and demonstrate OMF block's effectiveness in capturing inter-modal synergetic interactions.
> ||ACC|
> |-|-|
> |*Single*|0.487/0.489|
> |*Union*|0.626|
> |*wo-$L_{OMF}$*|0.725|
> |*w-$L_{OMF}$*|0.812|
>
> **Q2.** As you suggested, we increased the training epochs from 100 to 1,600 and observed that our method slightly improved, with macro-accuracy increasing from 63.6% to 65.4%. We agree that introducing KS or VGGSound datasets can help to more accurately evaluate our method. However, due to the limited rebuttal period and the very large data sizes involved, we plan to leave these results in the revised manuscript later.
>
> **Questions For Authors:**
>
> **Q1.** Thank you for pointing us to these recent methods. Since OGM-GE is originally designed for the classification task, we have included it as a benchmark for CREMA-D only (note that CMU-MOSI is a regression task, and anomalous tissue detection is an SVDD-based unsupervised task). Specifically, OGM-GE achieves a 61.0% accuracy on CREMA-D. Regarding MLA, we regret that its codes have been withdrawn, preventing us from adding it as a benchmark. Nonetheless, we have now discussed both methods in the Related Work and Experiment sections of this revision.
>
> **Q2.** Concatenation can be viewed as a generalized addition for combining the representation $z$ and noise $e$. To see this point, for concatenation, we have $[z^T,e^T] [\matrix{ W_1 \cr W_2}] =z^TW_1+e^TW_2$, which reduces to additive form $(z+e)^TW$ when $W_1= W_2$. Notably, conditional GAN (Mirza et al. 2014) also integrates noise vectors via concatenation for controlled generation. Moreover, using concatenation allows us to replace $e$ with the fused MIB $\xi$ during main training without modifying TRB's network architecture, thus offering more flexible cross-modal interactions compared to addition. Our ablation study on the CREMA-D dataset further confirms that the concatenation of $e$ (or $\xi$) with $z$ slightly outperforms a simple additive combination (macro-accuracy: 63.6\% vs. 62.4\%).
>
> **Other Comments Or Suggestions:**
>
> Properties 1 has been removed from the main text.

---

### Official Review · Reviewer_rM9B · 2025-03-10

**Overall Recommendation:** 3

**Summary:**

The paper proposes the OMIB framework to learn optimal multimodal information bottleneck (MIB) representations. It introduces a theoretically grounded objective that sets the regularization weight ($\beta$) within a derived bound and dynamically adjusts weights per modality (using parameter $r$) to balance imbalanced task-relevant information. OMIB combines modality-specific encoders with an optimal multimodal fusion (OMF) block that uses cross-attention, and it is implemented via a variational approximation. Experiments on synthetic data and other tasks (emotion recognition, sentiment analysis, and anomalous tissue detection) are conducted.

**Claims And Evidence:**

The authors support their claims with both theoretical proofs (for weight bounds and dynamic adjustment) and extensive experiments. Synthetic data validates the optimality of $β$ (Figure 3), and downstream experiments show consistent performance gains (e.g., +11.4% AUC improvement in anomalous tissue detection). However, some gains are modest, for instance, sentiment analysis on the CMU-MOSI dataset.

**Essential References Not Discussed:**

The paper doesn't cite [1], which tackles a similar challenge but in a broader self-supervised learning context. That work defines optimal shared and modality-specific information, enabling the learning of disentangled multimodal representation spaces. It develops a theoretical framework to assess the quality of disentanglement, even in scenarios where the Minimum Necessary Information (MNI) cannot be achieved—a situation common in real-world applications.

[1] https://openreview.net/pdf?id=3n4RY25UWP

**Experimental Designs Or Analyses:**

The experimental designs are generally sound. The paper evaluates OMIB on synthetic datasets to verify theoretical claims and on multiple real-world supervised tasks for empirical validation. The use of ablation studies (Table 6) to assess the impact of the warm-up phase, cross-attention, and the dynamic regularization factor is a strong point.

**Methods And Evaluation Criteria:**

The proposed methods and evaluation criteria are appropriate. The framework is designed to address key MIB challenges (sufficiency vs. conciseness and modality imbalance), and the use of standard metrics (classification accuracy, AUC, F1) on diverse datasets (CREMA-D, CMU-MOSI, 10x-hBC) is well justified.

**Other Comments Or Suggestions:**

N/A

**Other Strengths And Weaknesses:**

Strengths:

- Introduces a rigorous theoretical foundation for dynamically balancing modality-specific information using a derived $\beta$ bound and parameter $r$.

- Validates the approach across synthetic and supervised tasks, demonstrating consistent performance improvements.

- Conducts ablation studies that clearly show the importance of key components (warm-up, cross-attention, OMF block).

Weaknesses:

- The theoretical bounds (e.g., $M_u$) reply on estimating $H_{v_i}$ and $I(v_i;v_j)$, which is non-trivial in practice. The paper briefly mentions using MINE but does not discuss robustness to estimation errors or scalability to high-dimensional data.

- While the cross-attention network’s $O(N⋅M^2)$ complexity is noted, its impact on training/inference time and scalability to large-scale datasets is unexplored. Comparisons with lighter fusion mechanisms (e.g., late fusion) would strengthen practicality claims.

- Modality Scalability: The extension to $>3$ modalities is mentioned but not empirically validated.

- The work focuses on supervised learning. Extensions of the OMIB framework to semi-supervised or self-supervised scenarios are especially very important given the current landscape in AI.

**Questions For Authors:**

N/A

**Relation To Broader Scientific Literature:**

The paper builds on existing work in multimodal fusion and the information bottleneck framework It addresses known limitations of ad hoc regularization by providing a theoretically derived bound and dynamic adjustment. This situates OMIB as a meaningful extension in the area.

**Theoretical Claims:**

I checked the correctness of the proofs presented for Propositions 1–2. The derivations appear logically sound based on standard information theory properties and variational approximations.

---

> ### Author Rebuttal · Authors · 2025-04-01
>
> **Weaknesses**
>
> **W1.** Thank you for reminding us of this important point. **Robustness to estimation errors**: MINE is a theoretically validated estimator with strong consistency, which is achieved by optimizing the Donsker-Varadhan (DV) representation, a lower bound of the true MI (Belghazi et al., 2018). The optimal function that tightens the DV bound can be approximated using neural networks owing to universal approximation theorems, ensuring MINE's convergence to the true MI as sample size grows. Our experiments use datasets with relatively large sample sizes (e.g., Pathological Tissue (10,129)), the estimation error thus is expected to be minor. To further mitigate batch-induced bias, we apply exponential moving averaging on gradients during MINE’s optimization. Additionally, in its original study, MINE was shown to accurately estimate mutual information for generating superior single-modal IB for MNIST classification, empirically demonstrating its reliability. Finally, the theoretical bounds $M_u$ and $M_l$ are designed conservatively to tolerate estimation errors. For instance, $M_u$ is set to $\frac{1}{3 (H(v_1)+H(v_2)-I(v_1;v_2))}$, which intentionally tightens the upper bound to ensure $\beta$ remains within a safe range even if $H(v)$or $I(v_i;v_j)$ are slightly misestimated, thereby safeguarding convergence to optimal MIB. **Scalability:** MINE scales linearly with data dimensionality and sample size, as shown in its original work. Moreover, its computational cost is amortized since it is only used once to estimate $M_l$ and $M_u$ prior to training.
>
> **W2.** We apologize for the confusion. Actually, OMIB adopts the late-fusion strategy (the OMF block) similar to L-MIB (Mai et al. 2022), where unimodal representations are first condensed to reduce noises via variational IB encoders before fusion using a light CAN. This contrasts with the heavier early fusion occurring in the full information space. To empirically verify OMF's scalability to large-scale datasets, we generated six large synthetic datasets as in SIM-I–III and measured OMF's training and inference time per epoch (see the table below). The results indicate that OMF scales linearly with large-scale datasets.
> ||CAN||
> |-|-|-|
> |Samples|Main Training Time per epoch (s)|Inference Time (s)|
> |1e+5|0.2|0.2|
> |2e+5|0.6|0.7|
> |4e+5|1.1|1.0|
> |6e+5|1.6|1.6|
> |8e+5|1.8|1.8|
> |1e+6|2.2|2.1|
>
> **W3.** Since most MIB studies consider up to three modalities, we followed this routine. To address your concern about modality scalability, we conducted an additional experiment on synthetic datasets with up to five modalities. Each modality consists of a shared task-relevant, a unique task-relevant, and a unique task-irrelevant component. To isolate the impact of the number of modalities, the dataset size was fixed at $10^5$. The observed time costs (see Table below) scale approximately quadratically with $M$, consistent with the expected $\mathcal{O}(M^2)$ complexity.
> |Modalities|Main Training Time (s)|Inference Time (s)|
> |-|-|-|
> |2|41|0.30|
> |3|82|0.64|
> |4|159|1.15|
> |5|267|1.88|
>
> **W4.** Thank you for this excellent comment, which coincides with the most significant challenge of deep MIB learning proposed by Shwartz-Ziv and LeCun (2023). Our primary goal is to establish the theoretical achievability of optimal MIB under the classical MIB paradigm, where downstream task labels are available. As noted by Tian et al. (2020), the optimal MIB inherently depends on the specific task, since what is relevant for one task may be irrelevant for another. So optimal MIB may not be well defined without task labels in the first place. In our formulation, both shared and Modality-Specific Task-Relevant (MSTR) contents are considered, which further complicates the extension to label-free settings. Regarding self-supervised learning (SSL), while recent methods (e.g., DISTANGLEDSSL) have attempted to learn optimal MIB representations without labels, they typically rely on the strong MultiView assumption (Sridharan et al. 2008) that neglects MSTR. When this assumption is violated, MSTR can be excluded. Although this issue can be mitigated by adding regularization to increase the MI between the representations and inputs, the achievability of task-specific optimal MIB is agnostic. Thus, the guarantee of achieving optimal MIB with SSL in the presence of MSTR could be unattainable. For semi-supervised learning, a potential approach is to train OMIB on the available labeled data and then propagate labels to unlabeled data, with regularization such as a prior label distribution from labeled data to enhance generalization. However, the achievability of optimal MIB is not guaranteed either.
>
> **Essential References Not Discussed**
>
> We also thank you for pointing us to the DISTANGLEDSSL paper, which offers valuable insights into the SSL-based MIB. We will include and discuss it in the related work section of the revised version.

---

### Official Review · Reviewer_Z1rd · 2025-03-18

**Overall Recommendation:** 3

**Summary:**

This paper proposes OMIB, a novel framework for learning optimal Mutual Information Bottleneck (MIB) representations in multimodal learning. The authors address the challenge of imbalanced task-relevant information across modalities, which is a key issue in multimodal fusion. OMIB employs a dynamically weighted regularization strategy to optimize mutual information while mitigating redundancy and preserving modality complementarity. The approach leverages VAEs, CAN, and a two-phase training strategy to ensure efficient and adaptive multimodal representation learning. Theoretical derivations establish the conditions for achieving optimal MIB, and the framework is validated on both synthetic and real-world datasets across multiple tasks.

**Claims And Evidence:**

The paper makes several key claims:
- OMIB achieves optimal MIB by dynamically balancing modality contributions using an adaptive weighting factor r. Supported by Proposition 2, which explicitly derives r and validates it with synthetic data experiments (SIM-{I-III}).
- OMIB effectively reduces redundancy and preserves complementary information across modalities. Validated through ablation studies, which show significant performance degradation when OMIB’s fusion mechanism (OMF) or CAN is removed.
- OMIB outperforms SOTA MIB-based and non-MIB-based fusion methods across multiple benchmarks. Empirical results on CREMA-D, CMU-MOSI, and tissue detection datasets .
- OMIB maintains computational efficiency and scalability. Complexity analysis confirms O(N) time complexity and scalability tests.

**Essential References Not Discussed:**

The related work section is thorough.

**Experimental Designs Or Analyses:**

Strengths: Synthetic Experiments (SIM-{I-III}), comparison against SOTA methods on real-world datasets, ablation studies and complexity analysis.

Weaknesses: No explicit handling of missing or corrupted modalities, no robustness analysis for biased datasets, generalization to unseen datasets or domains is not tested.

**Methods And Evaluation Criteria:**

Yes, however, the datasets chosen (CREMA-D, CMU-MOSI, tissue detection) are strong, but not the hardest multimodal learning benchmarks.

**Other Comments Or Suggestions:**

None.

**Other Strengths And Weaknesses:**

Strengths:
+ Avoids direct MI computation by reformulating the learning process using VAEs and CAN, making optimization more tractable.
+ Adaptive modality weighting via r-regularization ensures balanced modality contributions, preventing over-reliance on a dominant modality.
+ Two-phase training strategy (warm-up + main training) reduces gradient conflicts and improves stability in learning task-relevant features.
+  Benchmark performance is strong, consistently surpassing state-of-the-art methods in multiple tasks.
+  Scalability is validated, confirming OMIB’s efficiency for large-scale multimodal datasets.

Weaknesses:
- Important challenges remain, including robustness to missing modalities, generalization to unseen data. See below.

**Questions For Authors:**

- The weighting mechanism depends on the KL-divergence ratio, which may still introduce instability in cases where one modality has significantly less information than the other. Can the authors comment on this?
- The Cross-Attention Network (CAN) enhances modality fusion by ensuring that complementary information is shared across modalities. While CAN improves fusion, it is unclear if it explicitly enforces diversity. A contrastive learning term could further ensure that different modalities contribute non-redundant information.
- In cases of completely missing modalities, CAN alone may not be sufficient without additional mechanisms such as modality imputation or self-supervised learning. The experiments assume all modalities are always available. Real-world multimodal settings often face missing, corrupted, or misaligned modalities (e.g., a failed camera or noisy audio in speech datasets).
- The chosen datasets (CREMA-D, CMU-MOSI, tissue detection) are strong benchmarks, but they are not the hardest challenges in multimodal learning. E.g. Evaluation on medical imaging datasets with severe class imbalance.

**Relation To Broader Scientific Literature:**

The related work section is thorough and well-referenced.

**Theoretical Claims:**

- The authors provide a rigorous information-theoretic foundation and mathematically derive an upper bound for the mutual information regularization parameter, β.
- Theoretical results ensure sufficiency, consistency, redundancy, complementarity, and specificity in learned multimodal representations.
- Proposition 3 validates the achievability of optimal MIB, but a more detailed analysis of robustness conditions (e.g., adversarial robustness) is missing.

---

> ### Author Rebuttal · Authors · 2025-04-01
>
> **Questions For Authors**
>
> **Q1.** This is a good point. To mitigate potential instability arising from extreme KL-divergence ratio ($KL_r$), we adopt several strategies. First, the raw $KL_r$ is not directly used; instead, the weight $r$ is computed as $1-tahn(\cdot)$ and bounded, thus preventing extreme values. Since the $tahn$ function is smooth and saturates, $r$ will not change abruptly in case of significant information imbalance (e.g., very large $KL_r$), thus promoting the training stability. Second, we will add a small constant $\epsilon$ to the denominator in the computation of $KL_r$ to avoid division by $0$ and enhance numeric stability. Third, the batch-averaged KL ratio (i.e. $\frac{1}{N}\sum_{n=1}^N r_n$) is used to compute $r$, thus smoothing out sample-level fluctuations of information ratio. Finally, Proposition 1 guarantees the convergence to the optimal MIB under bounded $r$ and $\beta$, regardless of the information imbalance. Empirical results from our synthetic experiments (SIM-I and -II), where one modality is designed to contain significantly more or less information than the other, further demonstrate this point.
>
> **Q2.** Thanks for this insightful comment. While CAN does not include an explicit contrastive loss, our training objective implicitly enforces diversity. As proved in Lemma 1, CAN's training objective function (Equation 17) ensures that both consistent (shared) and diverse (modality-specific) task-relevant contents are captured in the MIB upon convergence. In particular, the redundancy penalty terms ($I(\xi;z_i)$) in the objective function prioritize diverse, informative content as modality‐specific features incur less redundancy compared to shared features. Empirically, our synthetic experiments (Table 2) show that our method significantly surpasses MIB that exclusively contains shared\& task-relevant content, and achieves performance comparable to the authentic optimal MIB. These results confirm the effectiveness of our OMIB framework in accounting for information diversity. We acknowledge that incorporating a contrastive learning term could further promote diversity. However, adding such a term would complicate the theoretical analysis of achieving the optimal MIB and might disrupt the delicate balance between minimizing redundancy and maximizing task relevance established by our current objective. We consider this an intriguing direction for future work, particularly in exploring whether contrastive loss and redundancy penalty are functionally equivalent in enforcing diversity.
>
> **Q3.** Thank you for highlighting this practical concern. The primary goal of this work is to establish a rigorous theoretical foundation for achieving optimal MIB under the classical MIB paradigm. To allow tractable analysis, we adopt simplified assumptions, including the availability of all modalities, which admittedly do not fully capture real-world complexities. However, these assumptions align with the spirit of Ali Rahimi’s NIPS 2017 Test of Time Award remark—*“Simple experiments, simple theorems are the building blocks that help us understand more complicated systems”*.
>
> Your suggestion of incorporating additional mechanisms for handling missing modalities is valuable. One promising approach would be to use modality-complete data to train an auxiliary VAE that maps one modality's observations to the variational parameters of the other. In cases of modality-incomplete data, the available modality could then approximate the variational representations of the missing one via the reparameterization trick prior to CAN-mediated fusion. However, integrating this component would alter the framework's architecture and training objectives and compromise its theoretical guarantees, deviating from our study's original focus. Rather, these extensions constitute our *de novo* future work with a more practical orientation.
>
> **Q4.** We conducted an additional experiment using the MM-IMDb dataset—a challenging text-visual dataset with severe class imbalance. Specifically, it consists of 25,959 sample pairs across 23 movie genres, with the largest class containing 13,967 samples and the smallest 338, representing a 41-fold imbalance. A stratified split was applied to form training (60%), validation (10%), and testing (30%) sets. We evaluated our method and nine benchmark methods using the macro F1-score (see Table below). The results demonstrate that our method outperforms the benchmarks in this more challenging setting.
>
> |Methods|Concat|BiGated|MISA|deep IB|MMIB-Cui|MMIB-Zhang|E-MIB|L-MIB|C-MIB|OMIB|
> |-|-|-|-|-|-|-|-|-|-|-|
> |F1-score|0.218|0.309|0.334|0.374|0.353|0.373|0.303|0.377|0.357|0.409|

---

### Decision · Program_Chairs · 2025-05-01

**Decision:**

Accept (poster)

**Comment:**

This work presents OMIB, a theoretically grounded framework for learning optimal Mutual Information Bottleneck representations in multimodal learning by addressing imbalances in task-relevant information across modalities. OMIB dynamically adjusts regularization weights per modality and uses a two-phase training strategy with VAEs and cross-attention-based fusion to optimize information retention and reduce redundancy. Experimental results on synthetic and real-world tasks demonstrate its effectiveness in learning efficient and adaptive multimodal representations.

The paper has several strength, the synthetic experiments and the comparison against SOTA methods on real-world datasets, ablation studies and complexity analysis are well done. The work introduces a rigorous theoretical foundation for dynamically balancing modality-specific information using a derived  bound and parameter. In future work , the authors could explicitly handle missing or corrupted modalities, perform a robustness analysis for biased datasets, and empirically validate the extension to more than 3 modalities.

Overall, this is an interesting work that is of interest to the conference.